# Dual Prototype-Enhanced Contrastive Framework for Class-Imbalanced Graph Domain Adaptation

**Xin Ma**[1][*], **Yifan Wang**[2][*], **Siyu Yi**[3], **Wei Ju**[1][†], **Junyu Luo**[4], **Yusheng Zhao**[4],
**Xiao Luo**[5], **Jiancheng Lv**[1]

[1]College of Computer Science, Sichuan University
[2]School of Information Technology & Management,
University of International Business and Economics
[3]College of Mathematics, Sichuan University [4]Peking University [5]UCLA
maxin88@stu.scu.edu.cn, juwei@scu.edu.cn

## Abstract

Graph transfer learning, especially in unsupervised domain adaptation, aims to transfer knowledge from a label-abundant source graph to an unlabeled target graph. However, most existing approaches overlook the common issue of label imbalance in the source domain, typically assuming a balanced label distribution that rarely holds in practice. Moreover, they face challenges arising from biased knowledge in the source graph and substantial domain distribution shifts. To remedy the above challenges, we propose a dual-branch prototype-enhanced contrastive framework for graph domain adaptation under a class-imbalanced scenario. Specifically, we introduce a dual-branch graph encoder to capture both local and global information, generating class-specific prototypes from a distilled anchor set. Then, a prototype-enhanced contrastive learning framework is introduced. On the one hand, we encourage class alignment between the two branches based on constructed prototypes to alleviate the bias introduced by class imbalance. On the other hand, we infer the pseudo-labels for the target domain and align sample pairs across domains that share similar semantics to reduce domain discrepancies. Experimental results show that our ImGDA outperforms the state-of-the-art methods across multiple datasets and settings. The code is available at: https://github.com/maxin88scu/ImGDA.

## 1 Introduction

Graph serves as a versatile data structure to represent complex relationships [12, 47] in a number of fields such as social networks [2], molecular biology [13, 55] and recommender systems [20, 44]. One fundamental task for graph-structured data is node classification, which endeavors to predict the category of each node within the graph and is widely applied in various applications, i.e., community detection [51], smart city [58] and knowledge graph [49]. Nevertheless, the effectiveness of this task heavily relies on label information, which is time-consuming and costly [9]. Graph transfer learning [19, 61] has emerged as an effective framework for tackling this problem by leveraging labeled information from a source graph to facilitate learning on an unlabeled target graph, significantly enhancing the model's ability to generalize on the target graph.

Actually, there are several approaches that apply graph transfer learning for domain adaptation in the node classification task. Thanks to the powerful capabilities of graph neural networks (GNNs), unsupervised graph domain adaptation focuses on reducing the distribution discrepancy between

---

[*]Equal Contributions
[†]Corresponding Author

39th Conference on Neural Information Processing Systems (NeurIPS 2025).

target and source graphs within the latent representation space induced by GNNs [23, 53, 56]. Distance-based methods minimize a divergence measure between their distributions to learn invariant representations, such as maximum mean discrepancy [34] and graph subtree discrepancy [45] to align domains. Adversarial-based methods incorporate a discriminator to distinguish between the two domains and produce invariant features to confound the discriminator [4, 32].

Despite the promising performance of graph transfer learning, it often relies on the unrealistic assumption that the source domain graph exhibits a balanced label distribution. In practice, however, real-world graphs usually exhibit long-tailed structures, where most classes contain only a few labeled nodes (tail classes) and a small number of classes dominate with many labeled samples (head classes) [11, 14], resulting in severe class imbalance. For example, in the NCI dataset, which consists of graphs of chemical compounds [40], only around 5% of compounds exhibit anti-cancer activity, with the overwhelming remainder are labeled as inactive. Therefore, this class-imbalanced issue inevitably leads to the knowledge extraction bias [59] in the source graph which in turn adversely affects the graph domain adaptation. This naturally spurs a question: *How can label semantics in the target graph be effectively inferred under domain shifts and severe source imbalance?*

However, designing an effective framework for graph domain adaptation under class imbalance remains challenging due to several critical obstacles: ❶ *How to sufficiently alleviate the class imbalance for extracting knowledge from the source graph?* The latent space formed by imbalanced training data is highly skewed, with minor subspaces being compressed by the dominant ones. This imbalance forces the model to focus primarily on the head class, which results in insufficient learning of the tail class to extract unbiased knowledge. ❷ *How to effectively reduce the domain discrepancy*

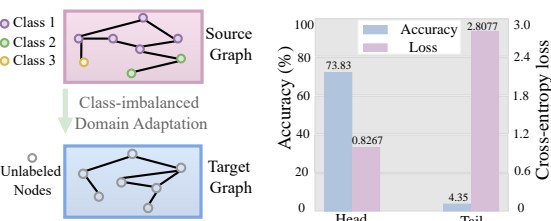

Figure 1: Illustration of class-imbalanced graph domain adaptation. The unsupervised domain adaptation method GDA-SpecReg suffers from an imbalanced class of source graphs, particularly for the tail class.

*to make accurate predictions on the target graph?* Since the target graph is entirely unlabeled, existing methods primarily focus on entire domain alignment, overlooking class-level distributions. Therefore, as the result shown in Figure 1, existing graph domain adaptation methods struggle to transfer the correct semantic knowledge, leading to poor performance, particularly for tail classes.

Towards this end, in this paper, we propose a holistic method termed **ImGDA**, a dual-branch prototype-enhanced contrastive framework for class-**Im**balanced **G**raph **D**omain **A**daption, which facilitates the transfer of class-imbalanced label information from the source graph to the unlabeled target graph. Specifically, we develop a dual graph encoder that jointly captures local and global information to learn generalized node representations. Building on this, a prototype for each class is generated from a distilled anchor set. Then, we introduce a prototype-aware contrastive learning framework that integrates cross-branch and cross-domain contrastive learning to enhance adaptation performance. To rebalance the feature space, the cross-branch prototype contrast encourages class alignment between the two branches. To further reduce domain discrepancy, we infer pseudo-labels of nodes from the target domain graph in a non-parametric manner and align sample pairs across domains that exhibit the same semantic meaning. Additionally, we treat temperature as a rebalancing parameter to mitigate class imbalance during training. Extensive experimental results validate the effectiveness of our proposed ImGDA for class-imbalanced graph domain adaptation.

## 2 Related Work

**Unsupervised Domain Adaptation (UDA).** UDA focuses on learning domain-invariant representations between a labeled source domain and an unlabeled target domain to transfer cross-domain information [24, 26, 43]. Recently, graph UDA methods have included adversarial learning and reducing domain discrepancies based on certain metrics (e.g., MMD [7], subtree discrepancy [45]). Among adversarial methods, RNA [25] adversarially extracts domain-invariant subgraphs to address domain shift, while leveraging spectral seriation for robust alignment under label scarcity. In the metric-based category, GDASpecReg [53] leverages spectral smoothness and maximum frequency response regularizations to enhance GNN transferability across node and link transfer scenarios,

while A2GNN [23] improves cross-domain transfer by adjusting GNN propagation layers using Lipschitz bounds. However, our ImGDA studies a novel yet practical scenario of imbalanced source data, and introduces a dual prototype-aware contrastive framework to reduce source graph bias and align the semantic space between source and target for cross-domain knowledge transfer.

**Class-Imbalanced Learning on Graphs.** It is well recognized that class imbalance on graphs can degrade classification performance. To address this problem, there are typically three approaches: (a) Modifying the loss function to prioritize underrepresented classes [3, 29, 36, 50]. (b) Post-hoc correction to modify logits for the tail class [10, 16]. (c) Re-sampling techniques that augment or generate tail class data [27, 31, 42, 57]. For instance, GraphSHA [22] expands tail class decision boundaries by creating more challenging samples, while ImGCL [54] addresses class imbalance by employing balanced sampling and explicitly accounting for node centrality within a contrastive learning paradigm. Extending these strategies to graph-level tasks, C$^3$GNN [14] addresses class-imbalanced graph classification by clustering majority classes and applying Mixup to learn hierarchical representations, while KDEX [28] transfers head-to-tail knowledge and trains diverse experts that are adaptively combined via a self-supervised router. Additionally, Qin et al. [33] further propose a comprehensive benchmark IGL-Bench for imbalanced graph learning, which evaluate the effectiveness, robustness, and efficiency of various algorithms within a unified framework. However, distribution shifts in graphs further complicate class-imbalanced learning on graphs. To address this, our proposed ImGDA introduces cross-branch prototype contrast to reduce class imbalance bias and cross-domain prototype contrast to align domain discrepancies, effectively generating domain-invariant representations.

## 3    Notations and Problem Definition

**Source Domain Graph.** Denote $\mathcal{G}^s = \{\mathcal{V}^s, \mathcal{E}^s, \boldsymbol{X}^s, \boldsymbol{Y}^s\}$ the source domain graph with the labeled node $\mathcal{V}^s$ and edge sets $\mathcal{E}^s$. The node feature matrix can be represented as $\boldsymbol{X}^s \in \mathbb{R}^{|\mathcal{V}^s| \times d}$, where entry $\boldsymbol{x}_v \in \mathbb{R}^d$ is associated with a feature vector of node $v$ with dimension $d$. We use the adjacency matrix $\boldsymbol{A}^s \in \mathbb{R}^{|\mathcal{V}^s| \times |\mathcal{V}^s|}$ to describe the structure information of the graph, where $\boldsymbol{A}^s_{ij} = 1$ if an edge exists between $v_i$ and $v_j$, i.e., $(v_i, v_j) \in \mathcal{E}^s$, otherwise, $\boldsymbol{A}^s_{ij} = 0$. The degree matrix is denoted as $\boldsymbol{D} = \mathrm{diag}(\boldsymbol{D}_1, \ldots, \boldsymbol{D}_N)$ with a degree of each node $\boldsymbol{D}_i = \sum_{j=1}^{|\mathcal{V}^s|} \boldsymbol{A}^s_{ij}$. We denote the node label matrix of the source domain graph as $\boldsymbol{Y}^s \in \mathbb{R}^{|\mathcal{V}^s| \times C}$, where $C$ corresponds to the total classes.

**Target Domain Graph.** The target domain graph is denoted as $\mathcal{G}^t = \{\mathcal{V}^t, \mathcal{E}^t, \boldsymbol{X}^t\}$ with unlabeled node set $\mathcal{V}^t$ and edge set $\mathcal{E}^t$. Similarly, the adjacency matrix $\boldsymbol{A}^t \in \mathbb{R}^{|\mathcal{V}^t| \times |\mathcal{V}^t|}$ indicates the node connectivity information in the target domain graph. And the feature matrix can be represented as $\boldsymbol{X}^t$. The attribute sets of the source and target domain graph could exhibit significant differences. Here, we construct a unified attribute set across both domains to align the dimensions.

**Problem Definition.** We consider $\mathcal{G}^s$ as a fully labeled source graph and $\mathcal{G}^t$ as an unlabeled target graph. Each $c$-th class in $\mathcal{G}^s$ contains $N_c$ nodes, ordered such that $N_C \geq N_2 \geq \cdots \geq N_C$. The source domain graph is assumed to be class-imbalanced, with the degree of imbalance quantified by the factor $N_1/N_C$. The objective of the task is to mitigate this class imbalance while transferring knowledge from the source graph to the target domain graph to achieve accurate node label prediction. Figure 2 provides a schematic overview of our proposed framework in the following.

## 4    Methodology

### 4.1    Dual-Branch Embedding Generalization

Generalized node embeddings are crucial for effective graph domain adaptation. Therefore, we introduce a dual-branch graph encoder to fully capture both local and global structure information of the graph [30, 62].

**Local Consistency Encoder.** To capture local consistency knowledge (i.e., neighboring samples are prone to share the same semantics), we directly utilize the Graph Convolutional Network (GCN) [18] as the encoder. Given the adjacency matrix $\boldsymbol{A}^*$ and feature matrix $\boldsymbol{X}^*$ ($* \in \{s, t\}$) of both source and target domain graphs, the output of the $l$-th layer is defined as:

$$\boldsymbol{Z}^{(l)}_{*,local} = \sigma(\tilde{\boldsymbol{D}}^{*-\frac{1}{2}} \tilde{\boldsymbol{A}}^* \tilde{\boldsymbol{D}}^{*-\frac{1}{2}} \boldsymbol{Z}^{(l-1)}_{*,local} \boldsymbol{W}^{*(l)}), \tag{1}$$

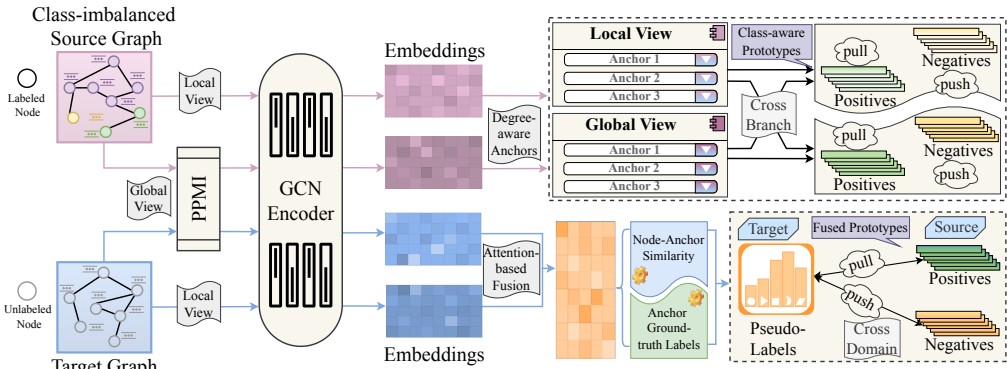

Figure 2: A schematic overview of our proposed ImGDA framework.

where $\tilde{\boldsymbol{A}}^* = \boldsymbol{I}_{|\mathcal{V}^*|} + \boldsymbol{A}^*$ is the adjacency matrix containing self-loop, $\tilde{\boldsymbol{D}}^*$ is the corresponding degree matrix accordingly. $\boldsymbol{W}^{*(l)}$ serves as the learnable filter in $l$-th layer and the initial feature matrix is $\boldsymbol{Z}_{*,local}^{(0)} = \boldsymbol{X}^*$. Here $\sigma(\cdot)$ denotes the activation function. By stacking $L$ graph convolutional layers, the extracted local consistency knowledge can be expressed as $\boldsymbol{Z}_{local}^* = \boldsymbol{Z}_{*,local}^{(L)}$.

**Global Consistency Encoder.** Furthermore, we utilize a graph encoding strategy based on positive pointwise mutual information (PPMI) [46, 62]. We represent the state at current time $t$ as $s(t) = v_i$, with the probability transit from the node $v_i$ to any of its neighbors $v_j$ expressed as:

$$\boldsymbol{P}_{ij}^* = p(s(t+1) = v_j | s(t) = v_i) = \boldsymbol{A}_{ij}^* / \boldsymbol{D}_i. \tag{2}$$

We apply the random walk guided by $\boldsymbol{P}^*$ to generate a collection of node paths on $\boldsymbol{A}^*$. From these paths, we construct co-occurrence frequency matrix $\boldsymbol{F}^* \in \mathbb{R}^{|\mathcal{V}^*| \times |\mathcal{V}^*|}$, where $\boldsymbol{F}_{ij}$ records how often node $v_j$ appears within a predefined window around $v_i$. The PPMI between nodes is calculated as:

$$\tilde{\boldsymbol{P}}_{ij}^* = \frac{\boldsymbol{P}_{ij}^*}{\sum_{i,j} \boldsymbol{P}_{ij}^*}, \tilde{\boldsymbol{P}}_i^* = \frac{\sum_j \boldsymbol{P}_{ij}^*}{\sum_{i,j} \boldsymbol{P}_{ij}^*}, \tilde{\boldsymbol{P}}_j^* = \frac{\sum_i \boldsymbol{P}_{ij}^*}{\sum_{i,j} \boldsymbol{P}_{ij}^*}, \quad \boldsymbol{M}_{ij}^* = \max\{\log(\frac{\tilde{\boldsymbol{P}}_{ij}^*}{\tilde{\boldsymbol{P}}_i^* \times \tilde{\boldsymbol{P}}_j^*}), 0\}, \tag{3}$$

where $\tilde{\boldsymbol{P}}_{ij}^*$ is the probability that $v_j$ appears in the $v_i$'s context, $\tilde{\boldsymbol{P}}_i^*$ and $\tilde{\boldsymbol{P}}_j^*$ are the estimated probability of node $v_i$ and context $v_j$ respectively. $\boldsymbol{M}_{ij}^*$ captures the high-order topological relationship between nodes. Thus, nodes that frequently co-occur at high frequency will have larger $\boldsymbol{M}_{ij}^*$ values compared to independent nodes. By treating the PPMI matrix $\boldsymbol{M}^*$ as a new adjacency matrix, we can effectively extract global consistency knowledge as:

$$\boldsymbol{Z}_{*,global}^{(l)} = \sigma(\boldsymbol{D}^{*-\frac{1}{2}} \boldsymbol{M}^* \boldsymbol{D}^{*-\frac{1}{2}} \boldsymbol{Z}_{*,global}^{(l-1)} \boldsymbol{W}^{*(l)}), \tag{4}$$

where $\boldsymbol{D}_i^* = \sum_j \boldsymbol{M}_{ij}^*$ and $\boldsymbol{W}^{*(l)}$ is shared learnable parameters used with the local consistency encoder. Similarly, the global structural consistency can be also extracted by stacking $L$ graph convolutional layers, namely, $\boldsymbol{Z}_{global}^* = \boldsymbol{Z}_{*,global}^{(L)}$.

**Attention-Based Consistency Fusion.** To fuse the extracted local and global consistency knowledge, we leverage the attention mechanism [38] and the attention coefficients can be obtained as:

$$\zeta_{ij} = \text{softmax}(\phi(\boldsymbol{W}^* \boldsymbol{z}_{i,local}^*, \boldsymbol{W}^* \boldsymbol{z}_{j,global}^*)), \tag{5}$$

where $\boldsymbol{W}^*$ is the shared parameter matrix, $\phi(\boldsymbol{z}_i, \boldsymbol{z}_j) = \text{LeakyRelu}(\boldsymbol{W}_0^{*\top}[\boldsymbol{z}_i || \boldsymbol{z}_j])$ denotes the attention function with parameter $\boldsymbol{W}_0^*$. The fused consistency knowledge of both domains can be:

$$\boldsymbol{Z}^* = \zeta_{ii} \boldsymbol{Z}_{local}^* + (1 - \zeta_{ii}) \boldsymbol{Z}_{global}^*. \tag{6}$$

## 4.2 Anchor-Based Prototype Construction

In contrast to directly using cross-entropy loss on class-imbalanced data, supervised contrastive learning (SCL) tends to achieve better results [15, 17, 60]. However, when the dataset is highly imbalanced, the feature space remains dominated by head class. To avoid overemphasis on head

classes, a straightforward approach is to generate a set of uniformly distributed prototypes for all classes, ensuring that each contributes approximately equally during optimization. The subsequent analysis provides theoretical guarantees that this strategy can effectively alleviate the class imbalance.

**Theorem 4.1.** *Let $\boldsymbol{Z} = \{\boldsymbol{z}_1, \ldots, \boldsymbol{z}_N\}, \|\boldsymbol{z}_i\| = 1$ be the extracted consistency knowledge of $N$ node points with label $\boldsymbol{Y} = \{y_1, \ldots, y_N\}$. The supervised contrastive loss for a class $c$ in a batch $\mathcal{B}$ is bounded by:*

$$\mathcal{L}_{SCL}(\boldsymbol{Z}; \boldsymbol{Y}, \mathcal{B}, c) \geq \sum_{i \in \mathcal{B}_c} \log((|\bar{\mathcal{B}}_c| - 1) + |\bar{\mathcal{B}}_c| \exp(\underbrace{\frac{1}{|\bar{\mathcal{B}}_c|} \sum_{k \in \bar{\mathcal{B}}_c} \boldsymbol{z}_i \cdot \boldsymbol{z}_k}_{repulsion\ term} - \underbrace{\frac{1}{|\mathcal{B}_c| - 1} \sum_{j \in \mathcal{B}_c \setminus \{i\}} \boldsymbol{z}_i \cdot \boldsymbol{z}_j}_{attraction\ term})), \quad (7)$$

*where $\mathcal{B}_c$ represents the subset within $\mathcal{B}$ containing all samples of class $c$ and $\bar{\mathcal{B}}_c$ denotes its complement set.*

The lower bound of SCL above is derived from [6] and comprises two terms. The attraction term encourages intra-class instances to converge toward their prototypes regardless of the class distribution, while the repulsion term enforces uniform inter-class separation and is dominated by head classes.

**Theorem 4.2.** *Let $\mathcal{C}_{\mathcal{B}}, |\mathcal{C}_{\mathcal{B}}| \leq C$ denotes the set of classes that appears in $\mathcal{B}$. The supervised contrastive loss for a class $c$ after class averaging is bounded by:*

$$\mathcal{L}_{SCL}(\boldsymbol{Z}; \boldsymbol{Y}, \mathcal{B}, c) \geq \sum_{i \in \mathcal{B}_c} \log(1 + (|\mathcal{C}_{\mathcal{B}}| - 1) \quad\quad\quad\quad\quad\quad (8)$$

$$\times \exp(\underbrace{\frac{1}{|\mathcal{C}_{\mathcal{B}}| - 1} \sum_{q \in \mathcal{C}_{\mathcal{B}} \setminus \{c\}} \frac{1}{|\mathcal{B}_q|} \sum_{k \in \mathcal{B}_q} \boldsymbol{z}_i \cdot \boldsymbol{z}_k}_{repulsion\ term} - \underbrace{\frac{1}{|\mathcal{B}_c| - 1} \sum_{j \in \mathcal{B}_c \setminus \{i\}} \boldsymbol{z}_i \cdot \boldsymbol{z}_j}_{attraction\ term})),$$

Therefore, head classes no longer dominate the repulsion term. To fully train all classes in $\mathcal{B}$, we learn the prototype of each class for prototype-aware contrastive learning. The proofs of the theorem are in Appendix A.

In practice, we distill the original graph to sample the anchor set consisting of the most important nodes. Then, we calculate the prototype based on the anchor set to facilitate effective learning. Specifically, for each class $c$ in the source domain graph, the top $k$ nodes ranked by the degree are selected to get the sub-graph of anchor nodes $\mathcal{G}_a^s$. We calculate the prototype as the mean vector of each class, namely, $\boldsymbol{\mu}_{c,q} = \frac{1}{k} \sum_{i=1}^{k} \boldsymbol{z}_{i,q}^s, y_i^s = c, q \in \{local, global\}$.

### 4.3 Prototype-Enhanced Contrastive Learning

Given the prototype sets of two branches for both source and target domain graphs, we formalize a prototype-aware contrastive learning framework, which integrates cross-branch and cross-domain prototype contrastive learning to mitigate class imbalance and effective domain adaptation.

**Cross-Branch Prototype Contrast.** Considering that the model extracts both the local and global consistency knowledge from both complementary branches, we contrast the consistency knowledge of the source domain graph between the two branches in a prototype manner to mitigate the imbalance effect [1]. Specifically, we calculate the prototype of both local and global consistency knowledge from two branches as $\{\boldsymbol{\mu}_{c,local}\}_{c=1}^{C}$ and $\{\boldsymbol{\mu}_{c,global}\}_{c=1}^{C}$. For each query node $v_i^s \in \mathcal{V}^s$, we use the prototype with the same class label as the positive sample and the cross-branch prototype contrastive loss can be defined as:

$$\mathcal{L}_{cb} = \frac{1}{4|\mathcal{V}^s|} \sum_{i=1}^{|\mathcal{V}_s|} \sum_{\boldsymbol{z}_+ = \boldsymbol{z}_{i,local}^s}^{\boldsymbol{z}_{i,global}^s} \sum_{\boldsymbol{\mu}_+ = \boldsymbol{\mu}_{y_i^s,local}}^{\boldsymbol{\mu}_{y_i^s,global}} \log\left(\frac{\exp(\boldsymbol{z}_+ \cdot \boldsymbol{\mu}_+/\tau)}{\exp(\boldsymbol{z}_+ \cdot \boldsymbol{\mu}_+/\tau) + \sum_{\boldsymbol{\mu} \in \mathcal{P}_i^-} \exp(\boldsymbol{z}_+ \cdot \boldsymbol{\mu}/\tau)}\right), \quad (9)$$

where $\mathcal{P}_i^-$ denote the prototype set excluding $\boldsymbol{u}_{y_i^s,q}, q \in \{local, global\}$ in two branches respectively, and $\tau$ is the temperature parameter for contrastive learning.

**Cross-Domain Prototype Contrast.** To alleviate the domain shift in the graph space, we seek to align the consistent knowledge of nodes in the source domain graph with nodes in the target domain graph that share the same semantics. To achieve this, we infer pseudo-labels of nodes from the target

domain graph in a non-parametric manner and employ cross-domain prototype contrastive learning between cross-domain pairs [52]. The pseudo-label for node $v_j^t$ in the target domain graph can be:

$$\hat{p}_j^t = \sum_{(v_i^s, y_i^s) \in \mathcal{G}_a^s} \Big( \frac{\exp(\boldsymbol{z}_j^t \cdot \boldsymbol{z}_i^s / \tau)}{\sum_{(v_i^s, y_i^s) \in \mathcal{G}_a^s} \exp(\boldsymbol{z}_j^t \cdot \boldsymbol{z}_i^s / \tau)} \Big) \boldsymbol{Y}_i^s, \tag{10}$$

where $\boldsymbol{Y}_i^s$ corresponds to the one-hot label of $v_i$. The pseudo-label can be derived as $\hat{y}_j^t = \arg\max(\hat{p}_j^t)$. Then, for each query node $v_j^t \in \mathcal{V}^t$, we pull semantically similar prototypes close compared to those with different semantics:

$$\mathcal{L}_{cd} = \frac{1}{|\mathcal{V}^t|} \sum_{j=1}^{|\mathcal{V}^t|} \log \Big( \frac{\exp(\boldsymbol{z}_j^t \cdot \boldsymbol{\mu}_{\hat{y}_j^t})}{\exp(\boldsymbol{z}_j^t \cdot \boldsymbol{\mu}_{\hat{y}_j^t}) + \sum_{\boldsymbol{\mu} \in \mathcal{P}_j^-} \exp(\boldsymbol{z}_j^t \cdot \boldsymbol{\mu})} \Big), \tag{11}$$

where $\mathcal{P}_j^-$ denotes the prototype set of fused consistency knowledge excluding $\boldsymbol{\mu}_{\hat{y}_j^t}$ in the source domain graph. Note that the cross-domain contrastive learning process can be interpreted as an Expectation Maximization (EM) scheme, where the aligned semantics between the source and target domain nodes are inferred in the E-step, and the log-likelihood of the nodes in the target domain graph is maximized in the M-step. The proof can be seen in Appendix B.

**Adaptive Temperature Formulation.** The temperature parameter $\tau$ in contrastive learning controls the penalty on hard negative samples [41]. However, for tail classes, where node samples are fewer, increasing $\tau$ has a negligible impact but reduces their gradients, worsening class imbalance. We propose an adaptive mechanism in which the temperature for each class is adjusted according to the number of samples within that class:

$$\tau_c = \gamma + (1 - \gamma) \cdot N_c / N_C, \tag{12}$$

where $\tau_c$ is the temperature of class $c$, and $\gamma$ here denotes the minimum value of temperature.

### 4.4 Overall Optimization

To further reduce the impact of class imbalance in the source domain graph, we employ a logit compensation strategy to correct the consistency knowledge [26, 29, 59], summarized as follows:

$$\mathcal{L}_{lc} = -\lambda_c \log \frac{\exp(\varphi_y(\boldsymbol{z}_i^s) + \delta_c)}{\sum_{c'=1}^C \exp(\varphi_{c'}(\boldsymbol{z}_i^s) + \delta_{c'})}, \tag{13}$$

where $\varphi(\cdot)$ is the classification function that outputs the logit for each label, $\lambda_y$ represents the contribution weight for class $y_i^s$, and $\delta_c$ denotes the compensation value for class $c$. Here, we have $\lambda_c = 1$ and $\delta_c = \log N_c$, following the previous work [29]. Finally, the objective can be:

$$\mathcal{L} = \mathcal{L}_{cb} + \mathcal{L}_{cd} + \mathcal{L}_{lc}. \tag{14}$$

Typically, the dynamic weight hyperparameters can be used to balance these losses during training, but we find in practice that a simple addition already yields effective results.

## 5 Experiment

### 5.1 Experimental Settings

**Benchmark Datasets.** This study conducts experiments on three publicly accessible network datasets from the ArnetMiner [37]: ACMv9 (A), Citationv1 (C), and DBLPv7 (D). These datasets are derived from different sources and cover distinct time periods: ACM (post-2010), Microsoft Academic Graph (pre-2008), and DBLP (2004-2008), resulting in diverse domain characteristics. All datasets model academic papers as nodes and construct undirected edges to represent citation relationships. To realistically simulate real-world class imbalance scenarios, we employ the operation proposed by [31] for preprocessing source domain data. This involves iteratively adjusting the number of nodes within classes in the source domain to achieve specified imbalance factors (IF). The imbalance factor is defined as $\rho = N_1 / N_C$, where class populations follow $N_1 \geq N_2 \geq \cdots \geq N_C$, with $N_c$ denoting the node count for class $c$ in the source graph. The experiments on more metrics of imbalanced distribution can be found in Appendix E.

Table 1: Results of methods across varying imbalance factors ($\rho$). Here A⇒C represents using A as the source graph and C as the target graph. Scores are reported as micro-average (%) and macro-average (%). The top-performing method is shown in **bold** with the runner-up is underlined.

| Methods | IF ($\rho$) | A⇒C Micro | A⇒C Macro | A⇒D Micro | A⇒D Macro | C⇒A Micro | C⇒A Macro | C⇒D Micro | C⇒D Macro | D⇒A Micro | D⇒A Macro | D⇒C Micro | D⇒C Macro |
|---|---|---|---|---|---|---|---|---|---|---|---|---|---|
| GCN | 10 | $66.01_{\pm1.15}$ | $60.90_{\pm2.14}$ | $61.21_{\pm0.57}$ | $53.20_{\pm0.71}$ | $58.25_{\pm1.22}$ | $56.52_{\pm2.87}$ | $63.39_{\pm0.32}$ | $58.38_{\pm1.27}$ | $55.47_{\pm0.39}$ | $52.80_{\pm0.86}$ | $61.68_{\pm0.38}$ | $58.35_{\pm0.68}$ |
| | 20 | $62.28_{\pm1.17}$ | $55.48_{\pm1.79}$ | $60.75_{\pm0.69}$ | $51.60_{\pm1.48}$ | $51.17_{\pm0.58}$ | $43.78_{\pm1.54}$ | $59.00_{\pm0.88}$ | $48.22_{\pm1.57}$ | $50.93_{\pm2.70}$ | $45.12_{\pm4.44}$ | $54.02_{\pm0.27}$ | $46.18_{\pm0.55}$ |
| | 50 | $53.07_{\pm3.39}$ | $42.41_{\pm3.32}$ | $56.26_{\pm2.21}$ | $43.35_{\pm3.53}$ | $44.38_{\pm1.12}$ | $32.20_{\pm2.51}$ | $53.06_{\pm0.55}$ | $37.09_{\pm1.18}$ | $41.86_{\pm1.31}$ | $31.11_{\pm1.50}$ | $42.28_{\pm3.28}$ | $32.05_{\pm3.30}$ |
| GAT | 10 | $66.48_{\pm0.72}$ | $58.37_{\pm2.74}$ | $61.58_{\pm0.75}$ | $51.44_{\pm1.02}$ | $55.10_{\pm2.47}$ | $51.54_{\pm4.12}$ | $60.87_{\pm1.00}$ | $52.99_{\pm1.34}$ | $56.18_{\pm0.31}$ | $53.11_{\pm0.80}$ | $61.64_{\pm0.23}$ | $58.22_{\pm0.53}$ |
| | 20 | $61.63_{\pm3.34}$ | $50.54_{\pm3.72}$ | $61.04_{\pm1.87}$ | $50.10_{\pm1.46}$ | $47.18_{\pm1.04}$ | $37.30_{\pm1.61}$ | $55.95_{\pm1.26}$ | $43.32_{\pm2.17}$ | $49.18_{\pm1.59}$ | $42.20_{\pm2.78}$ | $53.05_{\pm3.25}$ | $45.29_{\pm2.63}$ |
| | 50 | $53.21_{\pm7.62}$ | $42.07_{\pm7.64}$ | $53.67_{\pm4.00}$ | $40.36_{\pm5.33}$ | $41.02_{\pm0.99}$ | $26.41_{\pm1.18}$ | $49.76_{\pm0.63}$ | $32.29_{\pm1.15}$ | $40.14_{\pm1.46}$ | $28.89_{\pm1.85}$ | $45.38_{\pm7.90}$ | $34.77_{\pm8.31}$ |
| GIN | 10 | $62.12_{\pm0.69}$ | $55.03_{\pm0.84}$ | $60.91_{\pm0.79}$ | $54.39_{\pm1.48}$ | $54.35_{\pm1.75}$ | $49.21_{\pm1.31}$ | $60.14_{\pm1.00}$ | $54.51_{\pm2.16}$ | $52.59_{\pm0.67}$ | $49.43_{\pm2.42}$ | $59.89_{\pm1.42}$ | $54.68_{\pm2.28}$ |
| | 20 | $60.03_{\pm1.03}$ | $50.98_{\pm0.94}$ | $59.12_{\pm1.05}$ | $48.27_{\pm0.92}$ | $50.38_{\pm2.75}$ | $42.80_{\pm3.88}$ | $55.87_{\pm2.47}$ | $46.42_{\pm4.71}$ | $47.79_{\pm1.49}$ | $41.02_{\pm2.77}$ | $53.47_{\pm2.49}$ | $44.83_{\pm2.61}$ |
| | 50 | $54.39_{\pm1.62}$ | $44.28_{\pm1.69}$ | $54.43_{\pm1.60}$ | $41.90_{\pm2.09}$ | $41.26_{\pm0.37}$ | $29.45_{\pm0.51}$ | $47.05_{\pm0.60}$ | $31.57_{\pm1.13}$ | $43.38_{\pm1.26}$ | $34.41_{\pm2.05}$ | $48.59_{\pm2.15}$ | $39.22_{\pm2.29}$ |
| GDA-SpecReg | 10 | $51.27_{\pm1.86}$ | $39.29_{\pm2.43}$ | $53.69_{\pm3.02}$ | $40.87_{\pm5.56}$ | $49.69_{\pm4.93}$ | $39.08_{\pm7.51}$ | $57.41_{\pm3.15}$ | $45.54_{\pm4.35}$ | $52.73_{\pm3.20}$ | $47.07_{\pm7.86}$ | $56.37_{\pm1.40}$ | $45.04_{\pm3.46}$ |
| | 20 | $49.68_{\pm1.42}$ | $36.44_{\pm1.70}$ | $51.34_{\pm3.70}$ | $35.82_{\pm3.00}$ | $48.11_{\pm4.07}$ | $38.20_{\pm6.83}$ | $51.64_{\pm1.96}$ | $35.15_{\pm3.49}$ | $43.84_{\pm3.79}$ | $31.89_{\pm3.72}$ | $44.53_{\pm6.48}$ | $31.21_{\pm6.80}$ |
| | 50 | $49.80_{\pm6.48}$ | $36.09_{\pm7.63}$ | $47.44_{\pm4.58}$ | $29.36_{\pm6.49}$ | $47.64_{\pm4.97}$ | $33.87_{\pm7.90}$ | $52.63_{\pm2.53}$ | $38.46_{\pm6.93}$ | $41.95_{\pm3.49}$ | $28.28_{\pm7.97}$ | $45.63_{\pm5.22}$ | $30.81_{\pm8.85}$ |
| A2GNN | 10 | $61.23_{\pm0.93}$ | $54.72_{\pm2.23}$ | $59.00_{\pm0.64}$ | $46.61_{\pm1.16}$ | $51.29_{\pm0.51}$ | $41.70_{\pm0.77}$ | $61.07_{\pm0.32}$ | $50.09_{\pm0.68}$ | $59.54_{\pm0.83}$ | $55.41_{\pm1.74}$ | $60.32_{\pm1.82}$ | $60.11_{\pm2.30}$ |
| | 20 | $35.95_{\pm0.99}$ | $25.28_{\pm1.45}$ | $49.06_{\pm1.28}$ | $35.11_{\pm1.11}$ | $46.09_{\pm0.19}$ | $33.40_{\pm0.39}$ | $56.91_{\pm0.24}$ | $42.06_{\pm0.38}$ | $41.31_{\pm0.79}$ | $31.44_{\pm1.20}$ | $41.00_{\pm0.70}$ | $31.44_{\pm1.04}$ |
| | 50 | $32.48_{\pm1.62}$ | $17.61_{\pm1.75}$ | $35.22_{\pm0.12}$ | $15.10_{\pm0.24}$ | $40.26_{\pm0.08}$ | $24.09_{\pm0.07}$ | $50.33_{\pm0.19}$ | $31.03_{\pm0.17}$ | $31.27_{\pm0.22}$ | $14.29_{\pm0.48}$ | $27.79_{\pm0.35}$ | $13.19_{\pm0.72}$ |
| GraphENS | 10 | $70.49_{\pm0.63}$ | $64.53_{\pm1.33}$ | $66.43_{\pm1.40}$ | $58.40_{\pm3.69}$ | $63.25_{\pm1.11}$ | $62.41_{\pm2.29}$ | $67.47_{\pm0.37}$ | $63.87_{\pm1.23}$ | $58.95_{\pm0.64}$ | $58.34_{\pm1.07}$ | $65.50_{\pm0.53}$ | $63.52_{\pm0.65}$ |
| | 20 | $68.12_{\pm1.26}$ | $57.59_{\pm1.13}$ | $64.41_{\pm0.70}$ | $53.09_{\pm1.11}$ | $59.61_{\pm1.30}$ | $51.96_{\pm1.31}$ | $65.58_{\pm1.02}$ | $57.31_{\pm3.39}$ | $54.72_{\pm0.46}$ | $49.54_{\pm1.89}$ | $61.11_{\pm2.69}$ | $52.47_{\pm1.99}$ |
| | 50 | $62.74_{\pm0.81}$ | $51.76_{\pm0.79}$ | $61.01_{\pm1.66}$ | $47.74_{\pm4.14}$ | $54.09_{\pm1.51}$ | $45.51_{\pm1.74}$ | $60.40_{\pm0.90}$ | $47.81_{\pm2.29}$ | $50.67_{\pm0.43}$ | $42.18_{\pm0.44}$ | $56.26_{\pm0.85}$ | $47.81_{\pm2.43}$ |
| TAM | 10 | $72.36_{\pm0.60}$ | $67.27_{\pm0.56}$ | $67.12_{\pm0.69}$ | $\underline{61.52}_{\pm1.05}$ | $64.49_{\pm0.41}$ | $63.40_{\pm1.34}$ | $\underline{70.10}_{\pm0.85}$ | $\underline{66.20}_{\pm1.02}$ | $59.17_{\pm0.68}$ | $55.36_{\pm1.29}$ | $66.65_{\pm0.39}$ | $63.16_{\pm0.77}$ |
| | 20 | $68.90_{\pm1.49}$ | $60.01_{\pm1.23}$ | $\underline{66.11}_{\pm1.90}$ | $\underline{56.88}_{\pm1.83}$ | $61.36_{\pm0.70}$ | $57.64_{\pm2.19}$ | $\underline{68.11}_{\pm0.69}$ | $\underline{62.57}_{\pm1.98}$ | $53.81_{\pm1.05}$ | $46.92_{\pm1.65}$ | $61.23_{\pm1.22}$ | $53.28_{\pm1.42}$ |
| | 50 | $64.93_{\pm3.74}$ | $\underline{55.77}_{\pm4.74}$ | $61.63_{\pm0.99}$ | $50.23_{\pm1.48}$ | $\underline{58.95}_{\pm1.64}$ | $50.99_{\pm2.06}$ | $64.66_{\pm1.19}$ | $\underline{54.71}_{\pm2.03}$ | $\underline{53.39}_{\pm1.61}$ | $\underline{44.93}_{\pm2.14}$ | $57.88_{\pm1.67}$ | $47.52_{\pm2.02}$ |
| GraphSHA | 10 | $\underline{73.20}_{\pm0.73}$ | $\underline{70.43}_{\pm1.14}$ | $61.79_{\pm0.09}$ | $56.09_{\pm1.47}$ | $\underline{66.07}_{\pm1.26}$ | $\underline{66.24}_{\pm1.34}$ | $63.05_{\pm0.93}$ | $56.69_{\pm4.64}$ | $\underline{63.28}_{\pm0.55}$ | $\underline{64.06}_{\pm0.54}$ | $\underline{71.11}_{\pm0.64}$ | $\underline{69.03}_{\pm0.64}$ |
| | 20 | $\underline{71.00}_{\pm1.58}$ | $\underline{64.33}_{\pm2.56}$ | $57.95_{\pm2.55}$ | $47.37_{\pm3.16}$ | $\underline{62.87}_{\pm1.74}$ | $60.41_{\pm2.91}$ | $62.72_{\pm3.73}$ | $53.91_{\pm4.00}$ | $\underline{57.56}_{\pm1.71}$ | $\underline{53.22}_{\pm4.22}$ | $\underline{66.93}_{\pm1.40}$ | $\underline{62.03}_{\pm1.82}$ |
| | 50 | $64.97_{\pm2.30}$ | $53.99_{\pm2.49}$ | $\underline{65.96}_{\pm1.44}$ | $\underline{54.07}_{\pm1.91}$ | $56.55_{\pm2.73}$ | $49.19_{\pm1.58}$ | $56.10_{\pm5.25}$ | $44.72_{\pm6.29}$ | $50.08_{\pm0.95}$ | $41.72_{\pm3.42}$ | $57.89_{\pm1.59}$ | $\underline{50.52}_{\pm2.26}$ |
| ImGDA (Ours) | 10 | $\mathbf{77.93}_{\pm1.03}$ | $\mathbf{73.98}_{\pm2.74}$ | $\mathbf{71.34}_{\pm2.61}$ | $\mathbf{66.82}_{\pm1.82}$ | $\mathbf{67.79}_{\pm0.62}$ | $\mathbf{68.22}_{\pm1.41}$ | $\mathbf{73.06}_{\pm2.16}$ | $\mathbf{69.62}_{\pm2.98}$ | $\mathbf{66.64}_{\pm1.33}$ | $\mathbf{66.86}_{\pm1.78}$ | $\mathbf{74.72}_{\pm0.55}$ | $\mathbf{72.86}_{\pm1.14}$ |
| | 20 | $\mathbf{77.30}_{\pm1.05}$ | $\mathbf{73.92}_{\pm0.83}$ | $\mathbf{71.23}_{\pm1.59}$ | $\mathbf{65.66}_{\pm1.25}$ | $\mathbf{67.28}_{\pm0.84}$ | $\mathbf{66.39}_{\pm1.33}$ | $\mathbf{68.48}_{\pm1.54}$ | $\mathbf{63.42}_{\pm2.03}$ | $\mathbf{63.92}_{\pm0.46}$ | $\mathbf{63.45}_{\pm0.74}$ | $\mathbf{71.99}_{\pm1.15}$ | $\mathbf{68.38}_{\pm2.50}$ |
| | 50 | $\mathbf{73.53}_{\pm1.40}$ | $\mathbf{67.42}_{\pm1.23}$ | $\mathbf{69.14}_{\pm1.63}$ | $\mathbf{61.27}_{\pm1.50}$ | $\mathbf{66.87}_{\pm1.71}$ | $\mathbf{65.45}_{\pm2.64}$ | $\mathbf{70.57}_{\pm1.19}$ | $\mathbf{66.32}_{\pm1.86}$ | $\mathbf{60.67}_{\pm0.98}$ | $\mathbf{58.96}_{\pm1.75}$ | $\mathbf{66.07}_{\pm0.96}$ | $\mathbf{62.77}_{\pm1.11}$ |

**Compared Baselines.** For our study on graph domain adaptation under class-imbalanced scenario, we employ three GNNs as baselines: GCN [18], GAT [39], and GIN [48]. Meanwhile, we incorporate two baselines that are specifically developed for domain adaptation: GDA-SpecReg [53] and A2GNN [23]. In addition, we include three methods recognized for their effectiveness in handling imbalanced graph learning: GraphENS [31], TAM [35], and GraphSHA [22].

**Implementation Details.** We conduct extensive experiments by alternately setting one domain as the source and the other two as targets. The source imbalance factor $\rho \in \{10, 20, 50\}$ covers mild to severe imbalance levels. For our ImGDA, we adopt GCN [18] as the backbone with a 512-dimensional feature space. The hyperparameters are set as: $\gamma = 0.5$, $K = 200$, and $\alpha = \beta = 1$. For a fair comparison, all baselines employ the same GNN encoder and are fine-tuned for optimal performance. Each method is evaluated on the target domain graph w.r.t. *micro-F1* and *macro-F1* scores, and the final results is averaged over five runs.

## 5.2 Performance Comparison

Table 1 presents our results for $\rho = \{10, 20, 50\}$, while additional results for $\rho = \{5, 100\}$ are provided in Appendix D.1. From the table, our ImGDA consistently achieves the best performance. And the comparison shows that our approach significantly outperforms all other methods in class-imbalanced domain adaptation tasks. Additionally, when using DBLP dataset as the source domain with fewer samples, baseline methods generally perform worse

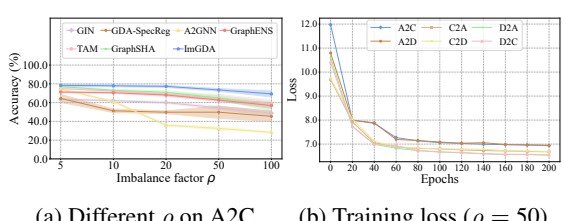

(a) Different $\rho$ on A2C    (b) Training loss ($\rho = 50$)

Figure 3: Impact of $\rho$ and convergence analysis.

than with other datasets. However, our ImGDA mitigates this issue by effectively capturing intrinsic semantics via a prototype-enhanced contrastive framework. Furthermore, Figure 3a shows the performance trend across different $\rho$ values in the A⇒C experiment, where $\rho$ ranges from 5 to 100. When $\rho$ is small (e.g., 5), performance gaps between methods are minor. As $\rho$ increases, all methods degrade due to rising imbalance. However, our ImGDA shows the smallest decline, maintaining stable results even under severe imbalance. Notably, as $\rho$ increases, domain adaptation-specific methods (GDA-SpecReg, A2GNN) drop sharply, while imbalance-handling methods (GraphENS, TAM, GraphSHA) outperform others. This indicates that addressing source-domain imbalance has a greater impact than mitigating domain shifts. Figure 3b also shows the rapid convergence of our method within a few epochs.

## 5.3 Ablation study

To evaluate the contribution of each component in our method, we conduct an ablation study. We remove three loss terms (w/o $\mathcal{L}_{lc}$, w/o $\mathcal{L}_{cb}$, w/o $\mathcal{L}_{cd}$), the global branch (w/o global, i.e., two branches are local branches), the local branch (w/o local, i.e., two branches are global branches), and the dynamic temperature (w/o $\gamma$), and assess the performance under $\rho = 50$. The results for the Micro score are shown in Figure 4, while the results for the

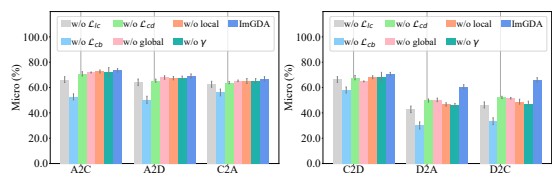

Figure 4: Results of ablation study on all data pairs ($\rho = 50$, Micro score (%) with standard deviation).

Macro score can be found in Appendix D.2. We can clearly see that excluding any component causes the performance degradation, especially the removal of $\mathcal{L}_{cb}$ (cross-branch prototype contrastive loss). This emphasizes the importance of addressing data imbalance in the source domain and enforcing consistency. Also, eliminating $\mathcal{L}_{cd}$ (cross-domain prototype contrastive loss) leads to some decline, but the impact is less severe, supporting the argument that mitigating data imbalance is more crucial than domain adaptation. Additionally, removing the $\mathcal{L}_{lc}$, global and local branches or switching to a fixed temperature reduces performance, particularly when DBLP is the source domain. The macro score drops after fixing the temperature emphasizes the importance of dynamic temperature in contrastive learning, where adjusting it based on class distribution alleviates data imbalance. These results validate the necessity and effectiveness of each component within our framework.

## 5.4 Sensitivity Study

**Effect of $k$.** We here study the impact of the number of anchor nodes $k$, where $k$ takes values from $\{50, 100, 200, 300, 400, 500\}$, as shown in Figure 5a. Both small and large values of $k$ degrade performance, with the impact more pronounced for small $k$. A small $k$ samples too few nodes, ignoring useful information and hindering learning. On the other hand, for a large $k$, while the number of head class nodes sampled remains unaffected, the number of tail class nodes sampled decreases due to the limited number of such nodes, leading to a data imbalance phenomenon similar to that

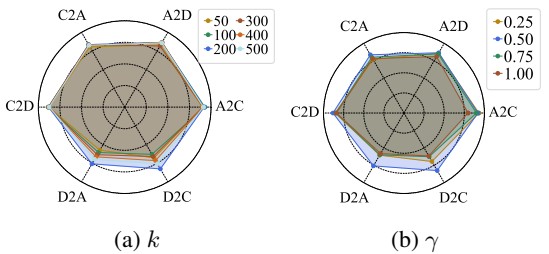

Figure 5: Impact of $k$ and $\gamma$ ($\rho = 50$). The distance between the points on each line and the center point represents the magnitude of the Micro score (%).

in the source domain, which also drops performance. Hence, selecting an appropriate number of anchor nodes $k$ is critical for effective prototype learning.

**Effect of $\gamma$.** We explore the impact of dynamic temperature $\gamma$ in Eq. (12), where $\gamma$ represents the minimum temperature associated with the tail class. It can be observed that as $\gamma$ decreases, the tail temperature lowers and the gradients increase, making $\gamma$ a weight parameter related to the gradients. We experiment with $\gamma$ in $\{0.25, 0.50, 0.75, 1.00\}$, as shown in Figure 5b. Compared to the default $\gamma = 0.5$, both high and low temperatures degrade performance, with lower temperatures slightly better. This highlights that smaller temperatures help alleviate data imbalance, but overly small values disproportionately increase the gradient of tail class, harming the training of other classes.

## 5.5 Capability to Mitigate the Imbalance Issue

We evaluate our ImGDA's capability to handle data imbalance by examining cross-entropy loss and accuracy across tail and head classes under $\rho = 20$. To enhance comparability, we normalize these metrics for each method and visualize the results in Figure 6. The results show that for head classes, all three methods achieve similar accuracy and cross-entropy loss ratios, indicating comparable effectiveness when abundant training data is available. However, for tail classes, the other two methods degrade significantly, revealing difficulty in handling imbalance. It can be observed that GraphENS performs better than GDA-SpecReg, likely due to its design advantages in dealing with imbalanced data. In comparison, our ImGDA consistently performs well in both the head and tail classes, underscoring its ability to mitigate class imbalance while ensuring robust generalization.

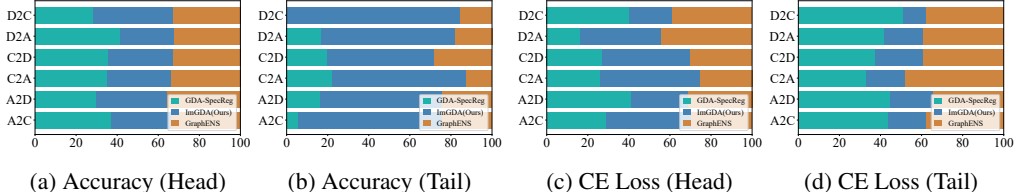

| (a) Accuracy (Head) | (b) Accuracy (Tail) | (c) CE Loss (Head) | (d) CE Loss (Tail) |

Figure 6: Comparison of head and tail class performance in terms of accuracy and cross-entropy (CE) loss. The horizontal axis here indicates the relative contribution of each compared method, normalized across all the three methods.

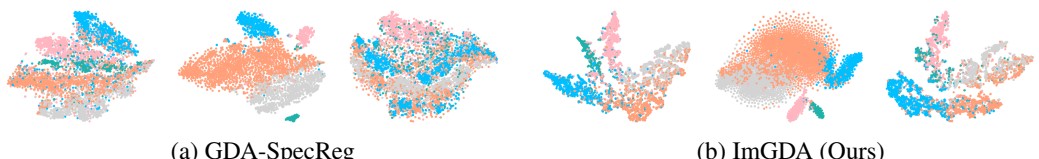

| (a) GDA-SpecReg | (b) ImGDA (Ours) |

Figure 8: *t*-SNE visualization of node embeddings from the baseline GDA-SpecReg and our proposed ImGDA. In each panel we sequentially present the balanced source domain graph, the imbalanced source domain graph, and the target domain graph, respectively.

## 5.6 Visualization

To further evaluate our ImGDA from a qualitative perspective, we compare its t-SNE visualizations with GDA-SpecReg. With $\rho$=20, node features learned by the encoder are projected into a 2D space to visualize three cases: a balanced source domain graph, an imbalanced source domain graph, and a target domain graph (Figure 8). As for the baseline, we can find that class boundaries appear blurred in both source domains, and the target domain shows a clear shift, indicating challenges in transferring domain-invariant features. In contrast, our ImGDA maintains well-separated class boundaries even though the source domain suffers from severe imbalance, and its target domain closely resembles that of the balanced source domain. This indicates that ImGDA effectively learns domain-invariant class structures and remains robust in the presence of imbalanced training data.

## 5.7 Different Sampling Strategies

To examine the impact of different anchor node sampling strategies on prototype computation, we compare three strategies: sampling based on node degree (Deg.), sampling based on inverse degree probability (Inv.), and random sampling (Rand.). We visualize the results for $\rho = 50$ in Figure 7, with additional results available in the Appendix D.4. From the figure, it can be observed that random sampling performs worse compared to the graph-structure-based sampling strategies, while the degree-based strategy employed by our method achieves the best performance. Additionally, our experiments reveal that inverse degree sampling significantly reduces training speed, further demonstrating the effectiveness of our selected sampling strategy.

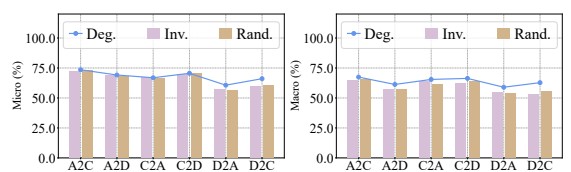

Figure 7: Results of sampling strategies ($\rho = 50$).

## 6 Conclusion

This paper tackles the challenge of class-imbalanced graphs by introducing a dual prototype-enhanced contrastive framework for graph domain adaptation. We use a dual graph encoder to capture local and global information and generate class-specific prototypes from a distilled anchor set. A prototype-aware contrastive learning module is introduced, combining cross-branch and cross-domain contrast. It mitigates source-domain imbalance via class alignment and reduces domain discrepancy by generating pseudo-labels to align semantically similar cross-domain pairs. Extensive experiments demonstrate the effectiveness of ImGDA compared to state-of-the-art methods.

## Acknowledgements

Jiancheng Lv and Xin Ma are supported in part by the National Major Scientific Instruments and Equipments Development Project of National Natural Science Foundation of China under Grant 62427820, the Fundamental Research Funds for the Central Universities under Grant 1082204112364. Wei Ju is supported by the National Natural Science Foundation of China under Grant 62306014, the Postdoctoral Fellowship Program (Grade A) of CPSF under Grant BX20250376, the Sichuan Science and Technology Program under Grant 2025ZNSFSC1506, the Fundamental Research Funds for the Central Universities under Grant 1082204112K97, and the Sichuan University Interdisciplinary Innovation Fund 1082204112J74. Siyu Yi is supported by the National Natural Science Foundation of China under Grant 12501344, the Postdoctoral Fellowship Program (Grade A) of CPSF under Grant BX20240239, the China Postdoctoral Science Foundation under Grant No. 2024M762201, and the Sichuan Science and Technology Program under Grant 2025ZNSFSC0808.

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

# A  Proof of Theorem 4.2

The supervised contrastive loss for a class $c$ in a batch $\mathcal{B}$ can be re-written as the following form [60]:

$$
\begin{aligned}
\mathcal{L}_{SCL}(\boldsymbol{Z}; \boldsymbol{Y}, \mathcal{B}, c) &= \sum_{i \in \mathcal{B}_c} -\frac{1}{|\mathcal{B}_c| - 1} \sum_{p \in \mathcal{B}_c \setminus \{i\}} \log \frac{\exp(\boldsymbol{z}_i \cdot \boldsymbol{z}_p)}{\sum\limits_{j \in \mathcal{Y}_B} \frac{1}{|\mathcal{B}_j|} \sum\limits_{k \in \mathcal{B}_j} \exp(\boldsymbol{z}_i \cdot \boldsymbol{z}_k)} \\
&= \sum_{i \in \mathcal{B}_c} \log \left( \frac{\sum\limits_{j \in \mathcal{C}_B} \frac{1}{|\mathcal{B}_j|} \sum\limits_{k \in \mathcal{B}_j} \exp(\boldsymbol{z}_i \cdot \boldsymbol{z}_k)}{\prod\limits_{p \in \mathcal{B}_c \setminus \{i\}} \exp(\boldsymbol{z}_i, \boldsymbol{z}_p)^{1/|\mathcal{B}_c| - 1}} \right) \\
&= \sum_{i \in \mathcal{B}_c} \log \left( \frac{\sum\limits_{j \in \mathcal{C}_B} \frac{1}{|\mathcal{B}_j|} \sum\limits_{k \in \mathcal{B}_j} \exp(\boldsymbol{z}_i \cdot \boldsymbol{z}_k)}{\exp\left(\frac{1}{|\mathcal{B}_c| - 1} \sum\limits_{p \in \mathcal{B}_c \setminus \{i\}} \boldsymbol{z}_i \cdot \boldsymbol{z}_p\right)} \right).
\end{aligned}
\tag{15}
$$

We divide the sum in the numerator into the positive and negative terms and since the exponential function is convex, we apply Jensen's inequality to get the lower bound of two terms as follows:

$$
\sum_{j \in \mathcal{C}_B} \frac{1}{|\mathcal{B}_j|} \sum_{k \in \mathcal{B}_j} \exp(\boldsymbol{z}_i \cdot \boldsymbol{z}_k) = \underbrace{\frac{1}{|\mathcal{B}_c| - 1} \sum_{k \in \mathcal{B}_c \setminus \{i\}} \exp(\boldsymbol{z}_i \cdot \boldsymbol{z}_k)}_{positive\ term} + \underbrace{\sum_{\substack{j \in \mathcal{C}_B \\ j \neq c}} \frac{1}{|\mathcal{B}_j|} \sum_{k \in \mathcal{B}_j} \exp(\boldsymbol{z}_i \cdot \boldsymbol{z}_k)}_{negative\ terms},
$$

$$
\frac{1}{|\mathcal{B}_c| - 1} \sum_{k \in \mathcal{B}_c \setminus \{i\}} \exp(\boldsymbol{z}_i \cdot \boldsymbol{z}_k) \geq \exp\left(\frac{1}{|\mathcal{B}_c| - 1} \sum_{k \in \mathcal{B}_c \setminus \{i\}} \boldsymbol{z}_i \cdot \boldsymbol{z}_k\right),
$$

$$
\sum_{\substack{j \in \mathcal{C}_B \\ j \neq c}} \frac{1}{|\mathcal{B}_j|} \sum_{k \in \mathcal{B}_j} \exp(\boldsymbol{z}_i \cdot \boldsymbol{z}_k) \geq \sum_{\substack{j \in \mathcal{C}_B \\ j \neq c}} \exp\left(\frac{1}{|\mathcal{B}_j|} \sum_{k \in \mathcal{B}_j} \boldsymbol{z}_i \cdot \boldsymbol{z}_k\right) \geq (|\mathcal{C}_B| - 1) \exp\left(\frac{1}{|\mathcal{C}_B| - 1} \sum_{\substack{j \in \mathcal{C}_B \\ j \neq c}} \frac{1}{|\mathcal{B}_j|} \sum_{k \in \mathcal{B}_j} \boldsymbol{z}_i \cdot \boldsymbol{z}_k\right).
\tag{16}
$$

And the lower bound of the numerator can be written as:

$$
\sum_{j \in \mathcal{C}_B} \frac{1}{|\mathcal{B}_j|} \sum_{k \in \mathcal{B}_j} \exp(\boldsymbol{z}_i \cdot \boldsymbol{z}_k) \geq \exp\left(\frac{1}{|\mathcal{B}_c| - 1} \sum_{k \in \mathcal{B}_c \setminus \{i\}} \boldsymbol{z}_i \cdot \boldsymbol{z}_k\right) + (|\mathcal{C}_B| - 1) \exp\left(\frac{1}{|\mathcal{C}_B| - 1} \sum_{\substack{j \in \mathcal{C}_B \\ j \neq c}} \frac{1}{|\mathcal{B}_j|} \sum_{k \in \mathcal{B}_j} \boldsymbol{z}_i \cdot \boldsymbol{z}_k\right).
\tag{17}
$$

Thus, the lower bound of supervised contrastive loss can be written as:

$$
\mathcal{L}_{SCL}(\boldsymbol{Z}; \boldsymbol{Y}, \mathcal{B}, c) \geq \sum_{i \in \mathcal{B}_c} \log\left(1 + (|\mathcal{C}_B| - 1) \exp\left(\underbrace{\frac{1}{|\mathcal{C}_B| - 1} \sum_{q \in \mathcal{C}_B \setminus \{c\}} \frac{1}{|\mathcal{B}_q|} \sum_{k \in \mathcal{B}_q} \boldsymbol{z}_i \cdot \boldsymbol{z}_k}_{repulsion\ term} - \underbrace{\frac{1}{|\mathcal{B}_c| - 1} \sum_{j \in \mathcal{B}_c \setminus \{i\}} \boldsymbol{z}_i \cdot \boldsymbol{z}_j}_{attraction\ term}\right)\right).
$$

$$\tag{18}$$

Thus, the proof of Theorem 4.2 is complete.

# B  Proof of Expectation-Maximization Perspective

In unsupervised graph domain adaptation, we aim to learn the graph encoder with parameter $\theta$ to maximize the log-likelihood of nodes in target graph $\mathcal{G}^t$ with source graphs $\mathcal{G}^s$, written as:

$$
\theta^* = \arg\max_{\theta} \sum_{v_j \in \mathcal{V}^t} \log \sum_{v_i = 1}^{\mathcal{V}^s} p(v_j^t, v_i^s; \theta).
\tag{19}
$$

We introduce a surrogate function $Q(v_i^s)$ ($\sum_{i=1}^{\mathcal{V}^s} Q(v_i^s) = 1$) to estimate the lower-bound via Jensen's inequality [21, 52]:

$$
\sum_{v_j^t \in \mathcal{V}^t} \log \sum_{i=1}^{\mathcal{V}^s} p(v_j^t, v_i^s; \theta) = \sum_{v_j^t \in \mathcal{V}^t} \log \sum_{i=1}^{\mathcal{V}^s} Q(v_i^s) \frac{p(v_j^t, v_i^s; \theta)}{Q(v_i^s)} \geq \sum_{v_j^t \in \mathcal{V}^t} \sum_{i=1}^{\mathcal{V}^s} Q(v_i^s) \log \frac{p(v_j^t, v_i^s; \theta)}{Q(v_i^s)}.
\tag{20}
$$

Note that the equality holds when $Q(v_i^s)/p(v_i^s, v_j^t; \theta)$ is constant. Thus, we have $Q(v_i^s) = p(v_i^s; v_j^t, \theta)$. Since $-\sum_{v_j^t \in \mathcal{G}^t} \sum_{i=1}^{\mathcal{G}^s} Q(v_i^s) \log Q(v_i^s)$ does not influence the optimization process, we objective can be re-written as:

$$\mathcal{L}_{cd} \geq \sum_{v_j^t \in \mathcal{V}^t} \sum_{i=1}^{\mathcal{V}^s} p(v_i^s; v_j^t, \theta) \log p(v_j^t, v_i^s; \theta). \tag{21}$$

Here we optimize the objective via an EM algorithm. In the E-step, we infer the posterior probability $p(v_i^s; v_j^t, \theta) = \frac{1}{|\Pi(j)|} \mathbb{I}(v_j^t, v_i^s)$ where indicator $\mathbb{I}(v_j^t, v_i^s) = 1$ if they has the same label and $|\Pi(j)| = \sum_{i=1}^{\mathcal{V}^s} \mathbb{I}(v_j^t, v_i^s)$. In the M-step, we aim to optimize the lower-bound, which can be defined as:

$$\theta = \arg\max_{\theta} \sum_{v_j^t \in \mathcal{V}^t} \frac{1}{|\Pi(j)|} \sum_{i=1}^{\mathcal{V}^s} \mathbb{I}(v_j^t, v_i^s) \log\Big(\frac{\exp(\boldsymbol{z}_j^t \cdot \boldsymbol{\mu}_{\hat{y}_j^t})}{\exp(\boldsymbol{z}_j^t \cdot \boldsymbol{\mu}_{\hat{y}_j^t}) + \sum_{\boldsymbol{\mu} \in \mathcal{P}_j^-} \exp(\boldsymbol{z}_j^t \cdot \boldsymbol{\mu})}\Big). \tag{22}$$

which is equivalent to our cross-domain contrastive learning objective in Eq. 11.

## C   Complexity Analysis

For the time complexity, let $N^s$ and $|\mathcal{E}^s|$ represent the number of nodes and edges in the source domain graph, and $|N^t|$ and $|\mathcal{E}^t|$ represent the number of nodes and edges in the target domain graph. Assuming an $L$-layer GCN encoder with a feature dimension of $d$, the computational complexity of feature encoding is $\mathcal{O}(L|\mathcal{E}|^s d + LN^s d^2)$ for the source domain, and $\mathcal{O}(L|\mathcal{E}|^t d + LN^t d^2)$ for the target domain. When $|\mathcal{E}|^s \gg n$, this simplifies to $\mathcal{O}(|\mathcal{E}|^s d)$ and $\mathcal{O}(|\mathcal{E}|^t d)$, respectively. For anchor node sampling, we sort nodes based on their degree and select $m$ nodes per class. Since $m \ll N^s$, the complexity is $\mathcal{O}(N^s \log N^s)$. For prototype computation, the Mean prototype computation per class takes $\mathcal{O}(md)$, the Cross-branch prototype contrastive loss takes $\mathcal{O}(md)$, the Cross-domain prototype contrastive loss takes $\mathcal{O}(N^t d)$ and the Supervised loss computation in the source domain takes $\mathcal{O}(N^s d)$. Thus, the total time complexity is $\mathcal{O}(|\mathcal{E}^s| d + |\mathcal{E}^t| d + N^s \log N^s + md + N^t d + N^s d)$, which can be approximated as: $\mathcal{O}(\max(|\mathcal{E}^s| + |\mathcal{E}^t|, N^s \log N^s, N^t d))$.

As for the space complexity: storing node features incurs a complexity of $\mathcal{O}(Nd)$; storing the sparse adjacency matrix requires $\mathcal{O}(|\mathcal{E}|)$; the PPMI matrix has a space complexity of $\mathcal{O}(N^2)$; the model parameters require $\mathcal{O}(D^2)$; and storing the prototypes and pseudo-labels requires $\mathcal{O}(Cd)$ and $\mathcal{O}(NC)$ respectively, and C denotes the number of classes. In summary, the main space cost of our method arises from the PPMI matrix, which is $\mathcal{O}(N^2)$.

## D   Supplement of Experiments

### D.1   Performance Experiment

To provide a more comprehensive evaluation of our ImGDA on class-imbalanced domain adaptation tasks, we consider additional scenarios beyond the main experiments presented in the body of the paper. Specifically, we explore more extreme settings with $\rho$ values in $\{5, 100\}$. All results are presented in Table 1 in the Appendix. At $\rho = 5$, the source graph is relatively less imbalanced, and the performance of all methods is similar, with generally good results. As $\rho$ increases, the advantages of our ImGDA become more pronounced. Even when $\rho = 100$, where the data is highly imbalanced, our ImGDA still maintains stable performance and significantly outperforms other competitive baselines. This highlights the robustness and effectiveness of our proposed ImGDA in addressing class-imbalanced domain adaptation tasks.

### D.2   Ablation Study

To better validate which parts of our method ImGDA are effective and how much they contribute to the improvement in model performance, we conduct ablation experiments based on the full experiments. Specifically, we evaluate the effects of the three loss terms ($\mathcal{L}_{lc}, \mathcal{L}_{cb}, \mathcal{L}_{cd}$), the two branches (local and global), and the dynamic temperature ($\gamma$). Figure 1 and Table 2 in the Appendix show the macro score performance from the ablation study, which follows a trend similar to that of the micro scores.

Table 1: The complete experiment results of approaches on class-imbalanced domain adaptation with $\rho$ range from 5 to 100. The best are highlighted in **bold** and the second-best are underlined.

| Methods | IF($\rho$) | A⇒C Micro | A⇒C Macro | A⇒D Micro | A⇒D Macro | C⇒A Micro | C⇒A Macro | C⇒D Micro | C⇒D Macro | D⇒A Micro | D⇒A Macro | D⇒C Micro | D⇒C Macro |
|---|---|---|---|---|---|---|---|---|---|---|---|---|---|
| GCN | 5 | $69.51_{\pm0.81}$ | $66.61_{\pm0.64}$ | $64.53_{\pm0.92}$ | $60.20_{\pm0.93}$ | $64.07_{\pm0.36}$ | $64.65_{\pm0.50}$ | $68.52_{\pm0.31}$ | $65.56_{\pm0.52}$ | $60.04_{\pm0.17}$ | $60.10_{\pm0.33}$ | $66.88_{\pm0.20}$ | $65.18_{\pm0.33}$ |
| | 10 | $66.01_{\pm1.15}$ | $60.90_{\pm2.14}$ | $61.21_{\pm0.57}$ | $53.20_{\pm0.71}$ | $58.25_{\pm1.22}$ | $56.52_{\pm2.87}$ | $63.39_{\pm0.32}$ | $58.38_{\pm1.27}$ | $55.47_{\pm0.39}$ | $52.80_{\pm0.86}$ | $61.68_{\pm0.38}$ | $58.35_{\pm0.68}$ |
| | 20 | $62.28_{\pm1.17}$ | $55.48_{\pm1.79}$ | $60.75_{\pm0.69}$ | $51.60_{\pm1.48}$ | $51.17_{\pm0.58}$ | $43.78_{\pm1.54}$ | $59.00_{\pm0.88}$ | $48.22_{\pm1.57}$ | $50.93_{\pm2.70}$ | $45.12_{\pm4.44}$ | $54.02_{\pm0.27}$ | $46.18_{\pm0.55}$ |
| | 50 | $53.07_{\pm3.39}$ | $42.41_{\pm3.32}$ | $56.26_{\pm2.21}$ | $43.35_{\pm3.53}$ | $44.38_{\pm1.12}$ | $32.20_{\pm2.51}$ | $53.06_{\pm0.55}$ | $37.09_{\pm1.18}$ | $41.86_{\pm1.31}$ | $31.11_{\pm1.50}$ | $42.28_{\pm3.28}$ | $32.05_{\pm3.30}$ |
| | 100 | $45.54_{\pm2.24}$ | $34.20_{\pm1.17}$ | $51.34_{\pm2.44}$ | $36.95_{\pm2.66}$ | $41.54_{\pm0.72}$ | $26.75_{\pm1.41}$ | $49.76_{\pm0.93}$ | $31.25_{\pm1.61}$ | $36.37_{\pm0.17}$ | $22.72_{\pm0.76}$ | $38.40_{\pm3.73}$ | $25.99_{\pm4.22}$ |
| GAT | 5 | $69.96_{\pm0.95}$ | $66.17_{\pm1.03}$ | $65.53_{\pm0.30}$ | $61.43_{\pm0.46}$ | $63.34_{\pm0.92}$ | $63.74_{\pm1.02}$ | $67.20_{\pm0.91}$ | $64.02_{\pm1.53}$ | $60.67_{\pm0.28}$ | $60.74_{\pm0.46}$ | $68.13_{\pm0.85}$ | $65.58_{\pm1.17}$ |
| | 10 | $66.48_{\pm0.72}$ | $58.37_{\pm2.74}$ | $61.58_{\pm0.75}$ | $51.44_{\pm1.02}$ | $55.10_{\pm2.47}$ | $51.54_{\pm4.12}$ | $60.87_{\pm1.00}$ | $52.99_{\pm1.34}$ | $56.18_{\pm0.31}$ | $53.11_{\pm0.80}$ | $61.64_{\pm0.23}$ | $58.22_{\pm0.53}$ |
| | 20 | $61.63_{\pm3.34}$ | $50.54_{\pm3.72}$ | $61.04_{\pm1.87}$ | $50.10_{\pm1.46}$ | $47.18_{\pm1.04}$ | $37.30_{\pm1.61}$ | $55.95_{\pm1.26}$ | $43.32_{\pm2.17}$ | $49.18_{\pm1.59}$ | $42.20_{\pm2.78}$ | $53.05_{\pm3.25}$ | $45.29_{\pm2.63}$ |
| | 50 | $53.21_{\pm7.62}$ | $42.07_{\pm7.64}$ | $53.67_{\pm4.00}$ | $40.36_{\pm5.33}$ | $41.02_{\pm0.99}$ | $26.41_{\pm1.18}$ | $49.76_{\pm0.63}$ | $32.29_{\pm1.15}$ | $40.14_{\pm1.46}$ | $28.89_{\pm1.85}$ | $45.38_{\pm7.90}$ | $34.77_{\pm8.31}$ |
| | 100 | $38.52_{\pm4.23}$ | $25.13_{\pm4.65}$ | $45.19_{\pm5.76}$ | $29.49_{\pm7.86}$ | $38.35_{\pm1.13}$ | $22.45_{\pm1.76}$ | $46.25_{\pm0.47}$ | $26.89_{\pm0.49}$ | $35.31_{\pm0.56}$ | $20.88_{\pm0.82}$ | $31.91_{\pm1.78}$ | $18.58_{\pm1.90}$ |
| GIN | 5 | $63.36_{\pm0.37}$ | $58.32_{\pm0.89}$ | $61.11_{\pm0.35}$ | $54.84_{\pm0.67}$ | $58.76_{\pm0.47}$ | $57.47_{\pm0.76}$ | $64.52_{\pm0.39}$ | $60.99_{\pm0.45}$ | $56.69_{\pm0.36}$ | $55.89_{\pm0.53}$ | $64.51_{\pm0.42}$ | $62.02_{\pm0.52}$ |
| | 10 | $62.12_{\pm0.69}$ | $55.03_{\pm0.84}$ | $60.91_{\pm0.79}$ | $54.39_{\pm1.48}$ | $54.35_{\pm1.75}$ | $49.21_{\pm1.31}$ | $60.14_{\pm1.00}$ | $54.51_{\pm2.16}$ | $52.59_{\pm0.67}$ | $49.43_{\pm2.42}$ | $59.89_{\pm1.42}$ | $54.68_{\pm2.28}$ |
| | 20 | $60.03_{\pm1.03}$ | $50.98_{\pm0.94}$ | $59.12_{\pm1.05}$ | $48.27_{\pm0.92}$ | $50.38_{\pm2.75}$ | $42.80_{\pm3.88}$ | $55.87_{\pm2.47}$ | $46.42_{\pm4.71}$ | $47.79_{\pm1.49}$ | $41.02_{\pm2.77}$ | $53.47_{\pm2.49}$ | $44.83_{\pm2.61}$ |
| | 50 | $54.39_{\pm1.62}$ | $44.28_{\pm1.69}$ | $54.43_{\pm1.60}$ | $41.90_{\pm2.09}$ | $41.26_{\pm0.37}$ | $29.45_{\pm0.51}$ | $47.05_{\pm0.60}$ | $31.57_{\pm1.13}$ | $43.38_{\pm1.26}$ | $34.41_{\pm2.05}$ | $48.59_{\pm2.15}$ | $39.22_{\pm2.29}$ |
| | 100 | $48.82_{\pm2.59}$ | $37.42_{\pm2.28}$ | $50.15_{\pm1.22}$ | $35.91_{\pm2.19}$ | $36.97_{\pm0.24}$ | $22.75_{\pm0.45}$ | $43.65_{\pm0.43}$ | $25.90_{\pm1.58}$ | $37.83_{\pm2.14}$ | $24.27_{\pm4.59}$ | $40.01_{\pm4.39}$ | $26.99_{\pm5.83}$ |
| A2GNN | 5 | $74.43_{\pm0.46}$ | $70.85_{\pm0.77}$ | $65.08_{\pm0.59}$ | $59.76_{\pm1.52}$ | $64.54_{\pm1.12}$ | $64.36_{\pm1.42}$ | $69.25_{\pm0.71}$ | $65.81_{\pm1.07}$ | $\mathbf{68.82_{\pm0.35}}$ | $\mathbf{70.45_{\pm0.44}}$ | $\mathbf{78.28_{\pm0.21}}$ | $\mathbf{76.70_{\pm0.36}}$ |
| | 10 | $61.23_{\pm0.93}$ | $54.72_{\pm2.23}$ | $59.00_{\pm0.64}$ | $46.61_{\pm1.16}$ | $51.29_{\pm0.51}$ | $41.70_{\pm0.77}$ | $61.07_{\pm0.32}$ | $50.09_{\pm0.68}$ | $59.54_{\pm0.83}$ | $55.41_{\pm1.74}$ | $60.32_{\pm1.82}$ | $60.11_{\pm2.30}$ |
| | 20 | $35.95_{\pm0.99}$ | $25.28_{\pm1.45}$ | $49.06_{\pm1.28}$ | $35.11_{\pm1.11}$ | $46.09_{\pm0.19}$ | $33.40_{\pm0.39}$ | $56.91_{\pm0.24}$ | $42.06_{\pm0.38}$ | $41.31_{\pm0.79}$ | $31.44_{\pm1.20}$ | $41.00_{\pm0.70}$ | $31.44_{\pm1.04}$ |
| | 50 | $32.48_{\pm1.62}$ | $17.61_{\pm1.75}$ | $35.22_{\pm0.12}$ | $15.10_{\pm0.24}$ | $40.26_{\pm0.08}$ | $24.09_{\pm0.07}$ | $50.33_{\pm0.19}$ | $31.03_{\pm0.17}$ | $31.27_{\pm0.22}$ | $14.29_{\pm0.48}$ | $27.79_{\pm0.35}$ | $13.19_{\pm0.72}$ |
| | 100 | $28.37_{\pm0.61}$ | $12.57_{\pm0.84}$ | $34.13_{\pm0.08}$ | $12.35_{\pm0.17}$ | $38.58_{\pm0.18}$ | $22.08_{\pm0.16}$ | $46.16_{\pm0.62}$ | $25.98_{\pm0.49}$ | $29.79_{\pm0.08}$ | $10.45_{\pm0.34}$ | $28.31_{\pm0.19}$ | $13.76_{\pm0.34}$ |
| GDA-SpecReg | 5 | $64.51_{\pm4.22}$ | $53.23_{\pm5.35}$ | $59.49_{\pm5.16}$ | $48.61_{\pm6.38}$ | $57.66_{\pm5.69}$ | $55.46_{\pm7.39}$ | $64.04_{\pm3.87}$ | $55.30_{\pm4.54}$ | $53.42_{\pm4.48}$ | $45.43_{\pm6.85}$ | $61.75_{\pm2.95}$ | $52.49_{\pm4.53}$ |
| | 10 | $51.27_{\pm1.86}$ | $39.29_{\pm2.43}$ | $53.69_{\pm3.02}$ | $40.87_{\pm5.56}$ | $49.69_{\pm4.93}$ | $39.08_{\pm7.51}$ | $57.41_{\pm3.15}$ | $45.54_{\pm4.35}$ | $52.73_{\pm3.20}$ | $47.07_{\pm7.86}$ | $56.37_{\pm1.40}$ | $45.04_{\pm3.46}$ |
| | 20 | $49.68_{\pm1.42}$ | $36.44_{\pm1.70}$ | $51.34_{\pm3.70}$ | $35.82_{\pm3.00}$ | $48.11_{\pm4.07}$ | $38.20_{\pm6.83}$ | $51.64_{\pm1.96}$ | $35.15_{\pm3.49}$ | $43.84_{\pm3.79}$ | $31.89_{\pm3.72}$ | $44.53_{\pm6.48}$ | $31.21_{\pm6.80}$ |
| | 50 | $49.80_{\pm6.48}$ | $36.09_{\pm7.63}$ | $47.44_{\pm4.58}$ | $29.36_{\pm6.49}$ | $47.64_{\pm4.97}$ | $33.87_{\pm7.90}$ | $52.63_{\pm2.53}$ | $38.46_{\pm6.93}$ | $41.95_{\pm3.49}$ | $28.28_{\pm7.97}$ | $45.63_{\pm5.22}$ | $30.81_{\pm8.85}$ |
| | 100 | $45.35_{\pm5.42}$ | $30.65_{\pm5.54}$ | $46.44_{\pm3.33}$ | $27.08_{\pm5.21}$ | $44.66_{\pm2.65}$ | $31.91_{\pm6.15}$ | $50.88_{\pm2.84}$ | $34.10_{\pm7.70}$ | $38.93_{\pm2.76}$ | $23.07_{\pm4.25}$ | $42.88_{\pm3.62}$ | $27.97_{\pm3.76}$ |
| GraphSHA | 5 | $\underline{77.01_{\pm0.37}}$ | $\underline{75.26_{\pm0.40}}$ | $67.36_{\pm1.33}$ | $62.90_{\pm0.49}$ | $\underline{69.15_{\pm0.46}}$ | $\underline{69.91_{\pm0.46}}$ | $68.03_{\pm0.26}$ | $65.54_{\pm0.34}$ | $65.69_{\pm0.19}$ | $66.61_{\pm0.35}$ | $74.10_{\pm0.20}$ | $72.89_{\pm0.35}$ |
| | 10 | $\underline{73.20_{\pm0.73}}$ | $\underline{70.43_{\pm1.14}}$ | $61.79_{\pm0.09}$ | $56.09_{\pm1.47}$ | $\underline{66.07_{\pm1.26}}$ | $\underline{66.24_{\pm1.34}}$ | $63.05_{\pm0.93}$ | $56.69_{\pm4.64}$ | $\underline{63.28_{\pm0.55}}$ | $64.06_{\pm0.54}$ | $\underline{71.11_{\pm0.64}}$ | $\underline{69.03_{\pm0.64}}$ |
| | 20 | $\underline{71.00_{\pm1.58}}$ | $64.33_{\pm2.56}$ | $57.95_{\pm2.55}$ | $47.37_{\pm3.16}$ | $\underline{62.87_{\pm1.74}}$ | $\underline{60.41_{\pm2.91}}$ | $62.72_{\pm3.73}$ | $53.91_{\pm4.00}$ | $\underline{57.56_{\pm1.71}}$ | $\underline{53.22_{\pm4.22}}$ | $\underline{66.93_{\pm1.40}}$ | $62.03_{\pm1.82}$ |
| | 50 | $64.97_{\pm2.30}$ | $53.99_{\pm2.49}$ | $\underline{65.96_{\pm1.44}}$ | $\underline{54.07_{\pm1.91}}$ | $56.55_{\pm2.73}$ | $49.19_{\pm1.58}$ | $56.10_{\pm5.25}$ | $44.72_{\pm6.29}$ | $50.08_{\pm0.95}$ | $41.72_{\pm3.42}$ | $\underline{57.89_{\pm1.59}}$ | $50.52_{\pm2.26}$ |
| | 100 | $56.13_{\pm4.86}$ | $44.73_{\pm4.47}$ | $\underline{60.87_{\pm3.26}}$ | $47.25_{\pm3.79}$ | $51.82_{\pm3.13}$ | $42.68_{\pm2.95}$ | $56.63_{\pm5.80}$ | $45.16_{\pm8.37}$ | $43.47_{\pm2.99}$ | $32.60_{\pm4.18}$ | $47.84_{\pm4.22}$ | $37.33_{\pm4.40}$ |
| GraphENS | 5 | $71.26_{\pm0.61}$ | $68.65_{\pm0.59}$ | $67.26_{\pm0.79}$ | $63.25_{\pm0.86}$ | $65.35_{\pm0.55}$ | $65.92_{\pm0.53}$ | $69.23_{\pm0.42}$ | $66.72_{\pm0.74}$ | $61.91_{\pm0.32}$ | $62.92_{\pm0.25}$ | $68.04_{\pm0.51}$ | $64.91_{\pm2.10}$ |
| | 10 | $70.49_{\pm0.63}$ | $64.53_{\pm1.33}$ | $66.43_{\pm1.40}$ | $58.40_{\pm3.69}$ | $63.25_{\pm1.11}$ | $62.41_{\pm2.29}$ | $67.47_{\pm0.37}$ | $63.87_{\pm1.23}$ | $58.95_{\pm0.64}$ | $58.34_{\pm1.07}$ | $65.50_{\pm0.53}$ | $63.52_{\pm0.65}$ |
| | 20 | $68.12_{\pm1.26}$ | $57.59_{\pm1.13}$ | $64.41_{\pm0.70}$ | $53.09_{\pm1.11}$ | $59.61_{\pm1.30}$ | $51.96_{\pm1.31}$ | $65.58_{\pm1.02}$ | $57.31_{\pm3.39}$ | $54.72_{\pm0.46}$ | $49.54_{\pm1.89}$ | $61.11_{\pm0.49}$ | $52.47_{\pm1.99}$ |
| | 50 | $62.74_{\pm0.81}$ | $51.76_{\pm0.79}$ | $61.01_{\pm1.66}$ | $47.74_{\pm4.14}$ | $54.09_{\pm1.51}$ | $45.51_{\pm1.74}$ | $60.40_{\pm0.90}$ | $47.81_{\pm2.29}$ | $50.67_{\pm0.43}$ | $42.18_{\pm0.44}$ | $56.26_{\pm0.85}$ | $47.81_{\pm2.43}$ |
| | 100 | $56.88_{\pm2.99}$ | $44.33_{\pm4.53}$ | $57.12_{\pm1.91}$ | $41.54_{\pm3.48}$ | $52.12_{\pm3.56}$ | $43.76_{\pm3.10}$ | $57.25_{\pm1.97}$ | $43.10_{\pm1.01}$ | $44.39_{\pm1.02}$ | $32.31_{\pm1.04}$ | $48.91_{\pm1.90}$ | $35.50_{\pm1.41}$ |
| TAM | 5 | $74.91_{\pm0.25}$ | $72.17_{\pm0.59}$ | $\underline{68.20_{\pm0.50}}$ | $\underline{64.37_{\pm0.46}}$ | $67.21_{\pm0.31}$ | $\underline{67.78_{\pm0.32}}$ | $\underline{71.83_{\pm0.29}}$ | $\mathbf{68.92_{\pm0.35}}$ | $63.39_{\pm0.12}$ | $63.19_{\pm0.25}$ | $71.45_{\pm0.16}$ | $69.83_{\pm0.23}$ |
| | 10 | $72.36_{\pm0.50}$ | $67.27_{\pm0.56}$ | $\underline{67.12_{\pm0.69}}$ | $\underline{61.52_{\pm1.05}}$ | $64.49_{\pm0.41}$ | $63.40_{\pm1.34}$ | $\underline{70.10_{\pm0.85}}$ | $\underline{66.20_{\pm1.02}}$ | $59.17_{\pm0.68}$ | $55.36_{\pm1.29}$ | $66.65_{\pm0.39}$ | $63.16_{\pm0.77}$ |
| | 20 | $68.90_{\pm1.49}$ | $60.01_{\pm1.23}$ | $\underline{66.11_{\pm0.99}}$ | $\underline{56.88_{\pm1.83}}$ | $61.36_{\pm0.70}$ | $57.64_{\pm2.19}$ | $\underline{68.11_{\pm1.69}}$ | $62.57_{\pm1.98}$ | $53.81_{\pm1.05}$ | $46.92_{\pm1.65}$ | $61.23_{\pm1.22}$ | $53.28_{\pm1.42}$ |
| | 50 | $64.93_{\pm3.74}$ | $\underline{55.77_{\pm4.74}}$ | $61.63_{\pm0.99}$ | $50.23_{\pm1.48}$ | $\underline{58.95_{\pm1.64}}$ | $50.99_{\pm2.06}$ | $\underline{64.66_{\pm1.19}}$ | $54.71_{\pm2.03}$ | $\underline{53.39_{\pm1.61}}$ | $\underline{44.93_{\pm2.14}}$ | $\underline{57.88_{\pm1.67}}$ | $\underline{47.52_{\pm2.02}}$ |
| | 100 | $\underline{59.08_{\pm3.61}}$ | $\underline{48.70_{\pm3.60}}$ | $59.81_{\pm1.83}$ | $\underline{47.58_{\pm2.18}}$ | $\underline{55.54_{\pm1.43}}$ | $\underline{46.82_{\pm1.57}}$ | $63.03_{\pm0.95}$ | $51.54_{\pm1.58}$ | $46.92_{\pm3.00}$ | $36.33_{\pm3.89}$ | $52.51_{\pm3.57}$ | $40.28_{\pm4.04}$ |
| ImGDA (Ours) | 5 | $\mathbf{78.35_{\pm1.48}}$ | $\mathbf{75.02_{\pm2.21}}$ | $\mathbf{72.08_{\pm1.86}}$ | $\mathbf{67.73_{\pm2.88}}$ | $\underline{67.49_{\pm2.19}}$ | $67.66_{\pm2.35}$ | $\mathbf{72.14_{\pm1.74}}$ | $\underline{68.66_{\pm2.83}}$ | $\underline{67.65_{\pm0.97}}$ | $\underline{68.43_{\pm1.30}}$ | $\underline{76.06_{\pm0.47}}$ | $\underline{74.50_{\pm0.43}}$ |
| | 10 | $\mathbf{77.93_{\pm1.03}}$ | $\mathbf{73.98_{\pm2.74}}$ | $\mathbf{71.34_{\pm2.61}}$ | $\mathbf{66.82_{\pm1.82}}$ | $\mathbf{67.79_{\pm0.62}}$ | $\mathbf{68.22_{\pm1.41}}$ | $\mathbf{73.06_{\pm2.16}}$ | $\mathbf{69.62_{\pm2.98}}$ | $\mathbf{66.64_{\pm1.33}}$ | $\mathbf{66.86_{\pm1.78}}$ | $\mathbf{74.72_{\pm0.55}}$ | $\mathbf{72.86_{\pm1.14}}$ |
| | 20 | $\mathbf{77.30_{\pm1.05}}$ | $\mathbf{73.92_{\pm0.83}}$ | $\mathbf{71.23_{\pm1.59}}$ | $\mathbf{65.66_{\pm1.25}}$ | $\mathbf{67.28_{\pm0.84}}$ | $\mathbf{66.39_{\pm1.33}}$ | $68.48_{\pm1.54}$ | $63.42_{\pm2.03}$ | $\mathbf{63.92_{\pm0.46}}$ | $\mathbf{63.45_{\pm0.74}}$ | $\mathbf{71.99_{\pm1.15}}$ | $\mathbf{68.38_{\pm2.50}}$ |
| | 50 | $\mathbf{73.53_{\pm1.40}}$ | $\mathbf{67.42_{\pm1.23}}$ | $\mathbf{69.14_{\pm1.63}}$ | $\mathbf{61.27_{\pm1.50}}$ | $\mathbf{66.87_{\pm1.71}}$ | $\mathbf{65.45_{\pm2.64}}$ | $\mathbf{70.57_{\pm1.19}}$ | $\mathbf{66.32_{\pm1.86}}$ | $\mathbf{60.67_{\pm0.98}}$ | $\mathbf{58.96_{\pm1.75}}$ | $\mathbf{66.07_{\pm0.96}}$ | $\mathbf{62.77_{\pm1.11}}$ |
| | 100 | $\mathbf{69.30_{\pm3.20}}$ | $\mathbf{58.80_{\pm1.87}}$ | $\mathbf{68.61_{\pm2.13}}$ | $\mathbf{56.79_{\pm1.75}}$ | $\mathbf{60.22_{\pm1.24}}$ | $\mathbf{51.79_{\pm1.97}}$ | $\mathbf{64.74_{\pm1.78}}$ | $\mathbf{52.82_{\pm2.60}}$ | $\mathbf{52.98_{\pm3.42}}$ | $\mathbf{45.64_{\pm2.63}}$ | $\mathbf{58.79_{\pm2.23}}$ | $\mathbf{48.52_{\pm0.86}}$ |

Generally, removing any part of the method leads to a certain degree of performance degradation. However, removing $\mathcal{L}_{cb}$ leads to significant drops, which clearly demonstrates the importance of utilizing source domain information and mitigating source domain data imbalance.

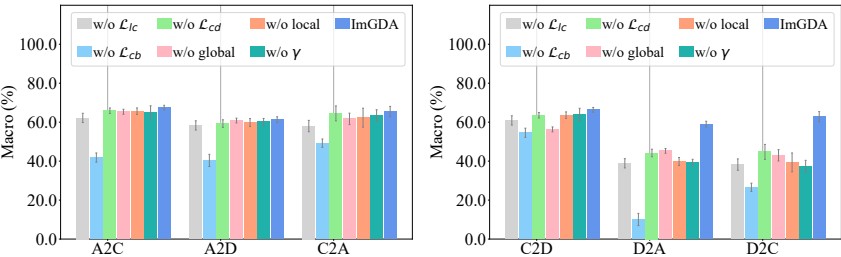

Figure 1: The results of ablation study ($\rho = 50$, Macro score (%) with standard deviation).

Table 2: The complete results of ablation study ($\rho = 50$).

| Methods | A⇒C Micro | A⇒C Macro | A⇒D Micro | A⇒D Macro | C⇒A Micro | C⇒A Macro | C⇒D Micro | C⇒D Macro | D⇒A Micro | D⇒A Macro | D⇒C Micro | D⇒C Macro |
|---|---|---|---|---|---|---|---|---|---|---|---|---|
| w/o $\mathcal{L}_{lc}$ | $66.28_{\pm2.27}$ | $62.17_{\pm2.38}$ | $64.16_{\pm2.59}$ | $58.33_{\pm2.38}$ | $62.58_{\pm2.38}$ | $58.00_{\pm2.92}$ | $66.35_{\pm2.04}$ | $60.89_{\pm2.71}$ | $42.79_{\pm2.44}$ | $38.88_{\pm2.10}$ | $46.27_{\pm2.73}$ | $38.23_{\pm2.22}$ |
| w/o $\mathcal{L}_{cb}$ | $52.63_{\pm2.40}$ | $41.84_{\pm2.35}$ | $50.33_{\pm2.83}$ | $40.36_{\pm3.12}$ | $56.12_{\pm2.70}$ | $49.23_{\pm2.14}$ | $58.09_{\pm2.78}$ | $54.56_{\pm2.37}$ | $30.04_{\pm0.96}$ | $10.06_{\pm1.86}$ | $33.43_{\pm1.90}$ | $26.54_{\pm1.01}$ |
| w/o $\mathcal{L}_{cd}$ | $70.81_{\pm1.71}$ | $65.87_{\pm1.40}$ | $65.44_{\pm1.22}$ | $59.27_{\pm1.94}$ | $63.90_{\pm0.87}$ | $64.50_{\pm3.83}$ | $67.81_{\pm0.74}$ | $63.54_{\pm3.06}$ | $49.75_{\pm1.94}$ | $44.17_{\pm2.53}$ | $52.22_{\pm2.06}$ | $44.75_{\pm2.53}$ |
| w/o global | $71.93_{\pm0.36}$ | $65.34_{\pm1.27}$ | $67.96_{\pm1.40}$ | $60.84_{\pm1.23}$ | $65.30_{\pm0.64}$ | $61.69_{\pm2.93}$ | $64.84_{\pm0.43}$ | $56.30_{\pm1.50}$ | $50.26_{\pm1.87}$ | $45.27_{\pm2.42}$ | $51.56_{\pm1.60}$ | $43.00_{\pm1.87}$ |
| w/o local | $72.69_{\pm1.17}$ | $65.63_{\pm1.67}$ | $67.40_{\pm1.33}$ | $59.80_{\pm2.04}$ | $65.15_{\pm1.95}$ | $62.30_{\pm4.84}$ | $68.14_{\pm0.96}$ | $63.60_{\pm2.60}$ | $46.69_{\pm1.43}$ | $39.82_{\pm1.98}$ | $48.84_{\pm1.52}$ | $39.34_{\pm1.19}$ |
| w/o $\gamma$ | $72.07_{\pm3.68}$ | $65.14_{\pm3.26}$ | $67.59_{\pm1.39}$ | $60.13_{\pm1.78}$ | $65.15_{\pm1.89}$ | $63.34_{\pm3.04}$ | $68.33_{\pm1.01}$ | $63.85_{\pm2.63}$ | $46.09_{\pm2.16}$ | $39.16_{\pm4.44}$ | $47.36_{\pm2.81}$ | $37.37_{\pm4.41}$ |
| ImGDA | $73.53_{\pm1.40}$ | $67.42_{\pm1.23}$ | $69.14_{\pm1.63}$ | $61.27_{\pm1.50}$ | $66.87_{\pm1.71}$ | $65.45_{\pm2.64}$ | $70.57_{\pm1.19}$ | $66.32_{\pm1.86}$ | $60.67_{\pm0.98}$ | $58.96_{\pm1.75}$ | $66.07_{\pm0.96}$ | $62.77_{\pm1.11}$ |

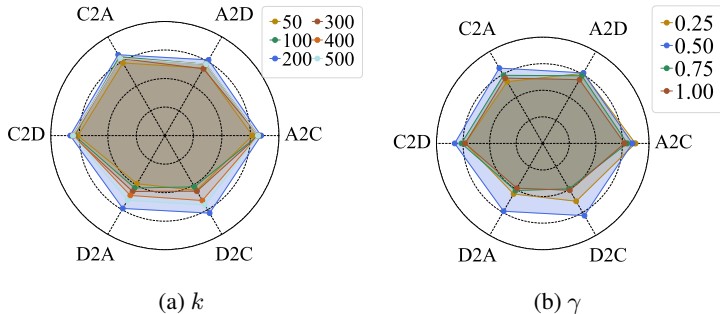

(a) $k$           (b) $\gamma$

Figure 2: Visualization of the sensitivity analysis of (a) $\gamma$ and (b) $k$. (Macro score (%))

## D.3 Sensitivity Study

In this section, we further present the sensitivity analysis results of two key hyperparameters concerning the Macro score, along with the complete numerical results for both metrics, to provide a clearer understanding of their impact on model performance. The experiments are conducted under the setting of $\rho = 50$.

Table 3: The impact of different values of $k$ on performance.

| $k$ | A⇒C Micro | A⇒C Macro | A⇒D Micro | A⇒D Macro | C⇒A Micro | C⇒A Macro | C⇒D Micro | C⇒D Macro | D⇒A Micro | D⇒A Macro | D⇒C Micro | D⇒C Macro |
|---|---|---|---|---|---|---|---|---|---|---|---|---|
| 50  | $72.11_{\pm1.44}$ | $61.40_{\pm1.56}$ | $66.38_{\pm2.86}$ | $54.64_{\pm2.61}$ | $64.42_{\pm1.30}$ | $58.83_{\pm2.80}$ | $70.45_{\pm1.55}$ | $60.99_{\pm3.10}$ | $46.00_{\pm3.17}$ | $38.97_{\pm2.33}$ | $54.34_{\pm3.25}$ | $44.39_{\pm2.50}$ |
| 100 | $74.18_{\pm1.72}$ | $65.36_{\pm1.59}$ | $68.89_{\pm1.33}$ | $57.77_{\pm0.68}$ | $66.10_{\pm1.96}$ | $63.84_{\pm2.38}$ | $70.79_{\pm0.72}$ | $63.23_{\pm2.96}$ | $48.55_{\pm2.76}$ | $42.40_{\pm2.07}$ | $50.53_{\pm2.55}$ | $41.50_{\pm3.08}$ |
| 200 | $73.53_{\pm1.40}$ | $67.42_{\pm1.23}$ | $69.14_{\pm1.63}$ | $61.27_{\pm1.50}$ | $66.87_{\pm1.71}$ | $65.45_{\pm2.64}$ | $70.57_{\pm1.19}$ | $66.32_{\pm1.86}$ | $60.67_{\pm0.98}$ | $58.96_{\pm1.75}$ | $66.07_{\pm0.96}$ | $62.77_{\pm1.11}$ |
| 300 | $72.44_{\pm1.58}$ | $65.63_{\pm3.14}$ | $65.35_{\pm3.02}$ | $54.08_{\pm2.52}$ | $66.72_{\pm1.23}$ | $62.44_{\pm2.24}$ | $70.50_{\pm1.00}$ | $64.62_{\pm3.86}$ | $50.89_{\pm1.62}$ | $45.24_{\pm2.84}$ | $52.84_{\pm1.70}$ | $45.00_{\pm2.67}$ |
| 400 | $72.33_{\pm1.43}$ | $65.45_{\pm0.97}$ | $69.32_{\pm0.93}$ | $57.78_{\pm2.30}$ | $65.93_{\pm2.20}$ | $62.83_{\pm2.09}$ | $70.31_{\pm0.77}$ | $63.70_{\pm3.86}$ | $53.18_{\pm1.82}$ | $48.45_{\pm2.99}$ | $57.24_{\pm3.66}$ | $52.56_{\pm2.11}$ |
| 500 | $72.96_{\pm2.02}$ | $65.35_{\pm0.78}$ | $68.64_{\pm1.67}$ | $57.85_{\pm1.34}$ | $66.12_{\pm1.65}$ | $62.68_{\pm2.37}$ | $69.92_{\pm0.95}$ | $64.39_{\pm3.44}$ | $55.80_{\pm2.44}$ | $52.32_{\pm3.77}$ | $60.59_{\pm3.02}$ | $56.12_{\pm2.74}$ |

**Effect of $k$.** To explore the impact of different values of $k$ (the number of anchor nodes) for our ImGDA, we conduct a sensitivity analysis of $k$. The results are shown in Table 3, and Figure 2a provides a visual analysis of the macro scores. It can be observed that both excessively small and large values of $k$ affect the model's performance. As discussed in Section 5.4 in the body of the paper, a too-small value of $k$ results in insufficient utilization of node information from the source domain, limiting the exploration of graph structural semantics. On the other hand, an excessively large $k$ may introduce data imbalance, leading to negative feedback that ultimately affects model performance.

**Effect of $\gamma$.** As discussed in Section 5.4 in the body of the paper, the hyperparameter $\gamma$ can be viewed as a weight parameter associated with the gradient of the tail class. To evaluate the impact of different values of $\gamma$ on model performance, we conduct a sensitivity analysis of $\gamma$. The results are shown in Table 4 in the Appendix, and Figure 2b provides a visual analysis of the macro scores. Both excessively small and large values of $\gamma$ lead to a decline in model performance, indicating that the gradient of the tail class should maintain a moderate weight during model training (i.e., too small a value affects the classification performance of other classes, while too large a value negatively impacts the classification of the tail class itself).

## D.4 Analysis of Different Sampling Strategies

Different sampling strategies affect the quality of the sampled anchor nodes. To investigate the impact of different sampling strategies on model performance, we conduct a comparative analysis of three different sampling strategies. Table 5 in the Appendix presents the experiment results with $\rho = \{20, 50, 100\}$. It can be observed that graph-structure-based sampling strategies (Deg. and Inv.) significantly outperform random sampling (Rand.), highlighting the importance of structural information in learning meaningful node representations. Furthermore, compared to the inverse degree-based strategy, our degree-based sampling approach achieves better performance, possibly because a node's local subgraph structure plays a crucial role in its representation within the entire graph. While the inverse degree-based strategy balances sampling bias, it inevitably leads to greater structural semantic loss by disregarding key graph information. Additionally, our experiments show

Table 4: The impact of different values of $\gamma$ on performance.

| $\gamma$ | A⇒C | | A⇒D | | C⇒A | | C⇒D | | D⇒A | | D⇒C | |
|---|---|---|---|---|---|---|---|---|---|---|---|---|
| | Micro | Macro | Micro | Macro | Micro | Macro | Micro | Macro | Micro | Macro | Micro | Macro |
| 0.25 | $74.47_{\pm2.25}$ | $69.70_{\pm1.58}$ | $68.59_{\pm1.59}$ | $60.95_{\pm1.66}$ | $61.46_{\pm3.43}$ | $53.64_{\pm2.06}$ | $68.38_{\pm2.55}$ | $58.34_{\pm2.44}$ | $48.40_{\pm3.12}$ | $43.94_{\pm3.75}$ | $55.64_{\pm2.64}$ | $50.26_{\pm2.69}$ |
| 0.50 | $73.53_{\pm1.40}$ | $67.42_{\pm1.23}$ | $69.14_{\pm1.63}$ | $61.27_{\pm1.50}$ | $66.87_{\pm1.71}$ | $65.45_{\pm2.64}$ | $70.57_{\pm1.19}$ | $66.32_{\pm1.86}$ | $60.67_{\pm0.98}$ | $58.96_{\pm1.75}$ | $66.07_{\pm0.96}$ | $62.77_{\pm1.11}$ |
| 0.75 | $72.06_{\pm3.40}$ | $63.59_{\pm3.07}$ | $67.97_{\pm3.31}$ | $58.79_{\pm2.52}$ | $64.00_{\pm1.58}$ | $59.17_{\pm3.26}$ | $67.60_{\pm1.68}$ | $61.21_{\pm3.69}$ | $48.05_{\pm2.90}$ | $41.01_{\pm4.15}$ | $50.12_{\pm2.21}$ | $39.42_{\pm3.41}$ |
| 1.00 | $63.76_{\pm2.19}$ | $61.46_{\pm2.23}$ | $64.76_{\pm3.17}$ | $55.37_{\pm2.38}$ | $62.48_{\pm2.07}$ | $56.74_{\pm3.06}$ | $67.18_{\pm1.57}$ | $58.46_{\pm2.25}$ | $46.55_{\pm2.33}$ | $38.92_{\pm2.91}$ | $49.28_{\pm3.50}$ | $40.73_{\pm3.52}$ |

that the inverse degree-based sampling strategy results in significantly longer training times. These factors collectively justify our choice of the degree-based sampling strategy.

Table 5: The results of different sampling strategies.

| Sampling Strategy $\rho=20$ | A⇒C | | A⇒D | | C⇒A | | C⇒D | | D⇒A | | D⇒C | |
|---|---|---|---|---|---|---|---|---|---|---|---|---|
| | Micro | Macro | Micro | Macro | Micro | Macro | Micro | Macro | Micro | Macro | Micro | Macro |
| Deg. | $77.30_{\pm1.05}$ | $73.92_{\pm0.83}$ | $71.23_{\pm1.59}$ | $65.66_{\pm1.25}$ | $67.28_{\pm0.84}$ | $66.39_{\pm1.33}$ | $68.48_{\pm1.54}$ | $63.42_{\pm2.03}$ | $63.92_{\pm0.46}$ | $63.45_{\pm0.74}$ | $71.99_{\pm1.15}$ | $68.38_{\pm2.50}$ |
| Inv. | $77.57_{\pm1.17}$ | $74.15_{\pm1.16}$ | $71.27_{\pm0.36}$ | $65.24_{\pm0.97}$ | $65.90_{\pm0.81}$ | $61.14_{\pm3.46}$ | $69.80_{\pm1.10}$ | $61.28_{\pm3.15}$ | $64.47_{\pm0.54}$ | $64.26_{\pm0.54}$ | $71.40_{\pm0.76}$ | $69.18_{\pm0.94}$ |
| Rand. | $76.86_{\pm0.39}$ | $72.71_{\pm0.67}$ | $71.44_{\pm0.57}$ | $65.11_{\pm0.67}$ | $66.15_{\pm1.22}$ | $61.51_{\pm3.77}$ | $69.67_{\pm2.25}$ | $62.13_{\pm2.40}$ | $63.68_{\pm1.51}$ | $63.17_{\pm1.84}$ | $71.70_{\pm1.11}$ | $68.93_{\pm0.80}$ |
| sampling method $\rho=50$ | A⇒C | | A⇒D | | C⇒A | | C⇒D | | D⇒A | | D⇒C | |
| | Micro | Macro | Micro | Macro | Micro | Macro | Micro | Macro | Micro | Macro | Micro | Macro |
| Deg. | $73.53_{\pm1.40}$ | $67.42_{\pm1.23}$ | $69.14_{\pm1.63}$ | $61.27_{\pm1.50}$ | $66.87_{\pm1.71}$ | $65.45_{\pm2.64}$ | $70.57_{\pm1.19}$ | $66.32_{\pm1.86}$ | $60.67_{\pm0.98}$ | $58.96_{\pm1.75}$ | $66.07_{\pm0.96}$ | $62.77_{\pm1.11}$ |
| Inv. | $72.40_{\pm1.82}$ | $64.91_{\pm1.62}$ | $68.39_{\pm1.91}$ | $56.99_{\pm3.48}$ | $66.99_{\pm1.52}$ | $63.81_{\pm2.64}$ | $69.89_{\pm1.01}$ | $62.22_{\pm2.24}$ | $57.09_{\pm1.90}$ | $54.46_{\pm3.45}$ | $59.86_{\pm2.07}$ | $53.23_{\pm3.37}$ |
| Rand. | $73.15_{\pm1.12}$ | $65.87_{\pm0.17}$ | $68.04_{\pm1.18}$ | $57.38_{\pm1.86}$ | $66.39_{\pm1.16}$ | $61.04_{\pm2.93}$ | $70.31_{\pm0.83}$ | $63.83_{\pm3.25}$ | $56.65_{\pm1.54}$ | $53.41_{\pm2.10}$ | $60.68_{\pm2.66}$ | $55.12_{\pm3.77}$ |
| sampling method $\rho=100$ | A⇒C | | A⇒D | | C⇒A | | C⇒D | | D⇒A | | D⇒C | |
| | Micro | Macro | Micro | Macro | Micro | Macro | Micro | Macro | Micro | Macro | Micro | Macro |
| Deg. | $69.30_{\pm3.20}$ | $58.80_{\pm1.87}$ | $68.61_{\pm2.13}$ | $56.79_{\pm1.75}$ | $60.22_{\pm1.24}$ | $51.79_{\pm1.97}$ | $64.74_{\pm1.78}$ | $52.82_{\pm2.60}$ | $52.98_{\pm3.42}$ | $45.64_{\pm2.63}$ | $58.79_{\pm2.23}$ | $48.52_{\pm0.86}$ |
| Inv. | $69.25_{\pm4.37}$ | $60.36_{\pm4.11}$ | $67.38_{\pm2.62}$ | $55.61_{\pm2.39}$ | $58.94_{\pm1.90}$ | $48.19_{\pm2.55}$ | $64.36_{\pm1.21}$ | $50.98_{\pm2.90}$ | $54.66_{\pm1.71}$ | $47.59_{\pm1.42}$ | $58.08_{\pm4.11}$ | $48.20_{\pm2.45}$ |
| Rand. | $69.25_{\pm4.68}$ | $57.04_{\pm3.45}$ | $67.54_{\pm3.00}$ | $56.54_{\pm2.11}$ | $59.32_{\pm1.96}$ | $48.61_{\pm2.63}$ | $64.82_{\pm1.53}$ | $51.93_{\pm2.93}$ | $54.88_{\pm1.97}$ | $47.61_{\pm1.47}$ | $57.30_{\pm2.14}$ | $46.23_{\pm2.36}$ |

## D.5 Analysis of Different GNN Encoders

To explore the effect of different encoders in our method, we replace our encoder from GCN to GraphSAGE [8], a widely used classic GNN encoder, and conduct experiments under $\rho = 50$. The results are shown in Table 6. It can be observed that in our method, GCN outperforms GraphSAGE, possibly because GraphSAGE samples only a subset of a target node's neighbors, which limits the exploration of local graph structures and leads to a performance drop. This highlights the importance of selecting an appropriate backbone encoder.

Table 6: The results of different GNN encoders.

| Encoder | A⇒C | | A⇒D | | C⇒A | | C⇒D | | D⇒A | | D⇒C | |
|---|---|---|---|---|---|---|---|---|---|---|---|---|
| | Micro | Macro | Micro | Macro | Micro | Macro | Micro | Macro | Micro | Macro | Micro | Macro |
| GCN | $73.53_{\pm1.40}$ | $67.42_{\pm1.23}$ | $69.14_{\pm1.63}$ | $61.27_{\pm1.50}$ | $66.87_{\pm1.71}$ | $65.45_{\pm2.64}$ | $70.57_{\pm1.19}$ | $66.32_{\pm1.86}$ | $60.67_{\pm0.98}$ | $58.96_{\pm1.75}$ | $66.07_{\pm0.96}$ | $62.77_{\pm1.11}$ |
| GraphSAGE | $62.28_{\pm1.77}$ | $56.77_{\pm2.12}$ | $60.13_{\pm1.21}$ | $50.03_{\pm2.14}$ | $53.06_{\pm1.44}$ | $50.80_{\pm0.91}$ | $59.68_{\pm1.36}$ | $50.61_{\pm1.40}$ | $48.22_{\pm1.85}$ | $44.50_{\pm2.03}$ | $52.59_{\pm1.44}$ | $46.38_{\pm1.98}$ |

Table 7: Detailed values of Standard Deviation, Mean/Median Ratio, Gini Coefficient with different imbalance factors of different Datasets.

| Methods | IF($\rho$) | $\sigma$ | $\epsilon$ | $\delta$ |
|---|---|---|---|---|
| ACMv9 | 5 | 628.79 | 1.17 | 0.31 |
| | 10 | 714.46 | 1.36 | 0.41 |
| | 20 | 762.39 | 1.66 | 0.49 |
| | 50 | 798.08 | 2.25 | 0.57 |
| | 100 | 814.47 | 2.92 | 0.62 |
| Citationv1 | 5 | 525.18 | 1.17 | 0.31 |
| | 10 | 596.31 | 1.36 | 0.41 |
| | 20 | 636.40 | 1.66 | 0.39 |
| | 50 | 666.23 | 2.25 | 0.57 |
| | 100 | 679.81 | 2.92 | 0.62 |
| DBLPv7 | 5 | 420.95 | 1.17 | 0.31 |
| | 10 | 471.26 | 1.36 | 0.41 |
| | 20 | 502.90 | 1.66 | 0.49 |
| | 50 | 526.27 | 2.25 | 0.57 |
| | 100 | 537.29 | 2.92 | 0.62 |

# E  Additional Metrics of Imbalanced Distributions

In our experiments, we employ the imbalance factor ($\rho$) to quantify the degree of class imbalance in the dataset, defined as:

$$\rho = \max\{N_1, N_2, \cdots, N_C\} / \min\{N_1, N_2, \cdots, N_C\}, \tag{23}$$

where $N_i$ denotes the number of nodes belonging to class $i$ in the source graph. To provide a more comprehensive assessment of imbalance across different datasets, we further introduce several additional metrics to evaluate the distributional skewness of the data.

- Standard Deviation ($\sigma$): It is widely used in probability and statistics to measure statistical dispersion, and in some cases it can reflect sampling uncertainty, which is defined as:

$$\sigma = \sqrt{\frac{1}{C} \sum_{i=1}^{C} (n_i - \bar{n}_i)^2}, \tag{24}$$

where $C$ represents the number of classes, $n_i$ represents the instance number of class $i$, and $\bar{n}_i$ represents the average number of instances.

- Mean/Median Ratio ($\epsilon$): The median is a fundamental statistical measure widely applied in fields such as economics, sociology, and medicine. Unlike the mean, it is less sensitive to extreme values and thus provides a more robust representation of the data distribution. Consequently, the mean-to-median ratio serves as an indicator of data skewness, defined as:

$$\epsilon = \frac{\text{mean}(N_1, N_2, \cdots, N_C)}{\text{median}(N_1, N_2, \cdots, N_C)}. \tag{25}$$

- Gini Coefficient ($\delta$): It was originally introduced by Italian economist Gini [5], measures distributional equality based on the Lorenz curve. Commonly used to quantify income or wealth inequality, it can similarly be applied as a metric of imbalanced distribution, as class imbalance parallels inequality across categories.

And we provide the corresponding Standard Deviation, Mean/Median Ratio, and Gini Coefficient values for each dataset under different imbalance factors in Table 7.

# F  Broader Impact and Limitations

Our method addresses class imbalance in graph domain adaptation and has the potential to benefit various applications such as scientific discovery, recommender systems, and bioinformatics by improving the representation of underrepresented classes. However, it also has limitations: the performance depends on the quality of the selected anchor set, the dual-branch design introduces additional computational overhead, and the pseudo-labeling process may suffer from noise under large domain shifts. Moreover, the framework requires the same label space across source and target domains, which could not be applied to open-set scenarios.

