# OpenReview forum: "Dual Prototype-Enhanced Contrastive Framework for Class-Imbalanced Graph Domain Adaptation"
_NeurIPS.cc/2025/Conference — NeurIPS 2025 poster_

### Official Review · Reviewer_XCp7 · 2025-06-14

**Clarity:** 2
**Significance:** 3
**Originality:** 3
**Rating:** 4
**Confidence:** 4

**Summary:**

This paper proposes a dual prototype-enhanced contrastive framework, ImGDA, which focuses on the challenge of class imbalance in graph domain adaptation. The goal is to transfer imbalanced label knowledge from a labeled source graph to an unlabeled target graph.

**Questions:**

1. In Section 3, the imbalance factor (IF) ρ is defined as N1/NC. What is the rationale behind this definition? When there are multiple tail classes, this formulation may not adequately reflect the degree of imbalance in the sample distribution. In the experimental setup, what is the ratio between head and tail classes?

2. The Cross-Domain Prototype Contrast relies on pseudo-labels in the target graph. How is the reliability of these pseudo-labels evaluated or ensured?

3. While Anchor-Based Prototype Construction may help alleviate class imbalance in the source graph, the Prototype-Enhanced Contrastive Learning still depends on pseudo-labels in the target graph. When the target domain also suffers from class imbalance, the pseudo-labels are likely to be biased toward head classes. Under such conditions, is the proposed method still robust and reliable?

**Ethical Concerns:**

["NO or VERY MINOR ethics concerns only"]

**Final Justification:**

The author clarified my concerns about the imbalance factor with a more complete computation method and relevant results. In addition, they offered a detailed explanation of the reliability of pseudo-labels and the experimental outcomes under a long-tailed distribution in the target domain. Their responses have addressed the main issues to a reasonable extent, and I find it appropriate to raise the score slightly.

**Limitations:**

The authors have discussed the limitations and potential negative societal impact of their work

**Quality:**

3

**Strengths And Weaknesses:**

Strengths:
This work is relatively innovative, with a well-organized and coherent structure.

Weaknesses:
Some technical details are insufficiently addressed, which impact the overall reliability of the proposed method.

---

> ### Author Rebuttal · Authors · 2025-07-28
>
> Thanks for your thoughtful suggestions. We will do our best to address your concerns below.
> >Q1: Explanation of Imbalance Factor.
>
> **A1:** Thank you for your comment. In our method, the imbalance factor is defined as $\rho = N_1 / N_c$, where $N_1$ and $N_c$ represent the number of samples in the head class and tail class, respectively. During preprocessing, we first compute a constant $\mu = \left(\frac{1}{\rho}\right)^{\frac{1}{C-1}}$, where $C$ denotes the total number of classes. Then, we calculate the number of nodes in each class using the formula: $N_i = N_{\text{max}} \times \mu^i$, where $N_{\text{max}}$ refers to the number of nodes in the most frequent class. This design ensures that the ratio between the head and tail classes is $\rho$, and the number of nodes in intermediate classes increases gradually from the tail class to the head class.
>
> This imbalance factor effectively captures the extremity of distributional skew in the dataset, which is crucial for long-tailed scenarios where head and tail classes exhibit typical behaviors: head classes often dominate and influence other categories, while tail classes are more susceptible to such influence. **By defining the imbalance in this way, we can better characterize and address the long-tailed nature of the data, which is widely used in imbalanced graph research[1][2][3][4] in recent years.**
>
> In our experiments, we set $\rho \in \\{5, 10, 20, 50, 100\\}$ to simulate various imbalance levels, ranging from mild to extreme, in the source domain. This design enables a thorough evaluation of the stability and effectiveness of our proposed method. We hope this explanation clarifies your concerns.
>
> [1] Park J, Song J, Yang E. Graphens: Neighbor-aware ego network synthesis for class-imbalanced node classification[C]//International conference on learning representations. 2021.
>
> [2]Li W Z, Wang C D, Xiong H, et al. Graphsha: Synthesizing harder samples for class-imbalanced node classification[C]//Proceedings of the 29th ACM SIGKDD conference on knowledge discovery and data mining. 2023: 1328-1340.
>
> [3]Song J, Park J, Yang E. TAM: topology-aware margin loss for class-imbalanced node classification[C]//International conference on machine learning. PMLR, 2022: 20369-20383.
>
> [4]Zhao T, Zhang X, Wang S. Graphsmote: Imbalanced node classification on graphs with graph neural networks[C]//Proceedings of the 14th ACM international conference on web search and data mining. 2021: 833-841.
> >Q2: Discussion about the Reliability of Pseudo-labels.
>
> **A2:** Thank you for your comment and for your attention to the technical details of our method. **Pseudo-label is widely used in several research[1][2][3][4],** and in our framework, we adopt a non-parametric approach to generate pseudo-labels for target domain nodes by measuring the embedding similarity between the source and target domains. Several design choices help ensure the quality of these pseudo-labels:
>
> - First, **the powerful representation learning capability of GNNs** enables effective feature extraction on graph-structured data. Nodes with the same label in the source and target domains often share similar neighborhood structures and features, resulting in embeddings that are highly comparable across domains.
>
> - Second, as shown in Equation (10), we compute the similarity between each target node and the anchor source nodes, and then use this similarity as a weight to aggregate the one-hot label encodings of the source nodes. The class with the highest aggregated score is selected as the pseudo-label. Unlike classifier-based labeling, this non-parametric method **leverages the rich label information from the source domain and considers multiple source nodes for each decision,** thereby enhancing pseudo-label reliability.
>
> - Third, we construct a **high-quality anchor set** by sampling nodes based on node degree. These anchors are then used for prototype construction and pseudo-label generation. Since high-degree nodes typically exhibit more representative category features and are less noisy, they help reduce the impact of label noise and further improve pseudo-label quality.
>
> In summary, our method incorporates several strategies to **enhance the reliability of pseudo-labels** in the target domain, which in turn supports high-quality contrastive learning and effective knowledge transfer. We hope this response addresses your concern.
>
> [1]Qiu Z, Zhang Y, Lin H, et al. Source-free domain adaptation via avatar prototype generation and adaptation[J]. arXiv preprint arXiv:2106.15326, 2021.
>
> [2]Qiao Z, Luo X, Xiao M, et al. Semi-supervised domain adaptation in graph transfer learning[J]. arXiv preprint arXiv:2309.10773, 2023.
>
>
> [3]Zhang Z, Liu M, Wang A, et al. Collaborate to adapt: Source-free graph domain adaptation via bi-directional adaptation[C]//Proceedings of the ACM Web Conference 2024. 2024: 664-675.
>
> [4]Chen N, Liu Z, Hooi B, et al. Consistency training with learnable data augmentation for graph anomaly detection with limited supervision[C]//The twelfth international conference on learning representations. 2024.
> >Q3: Discussion about the Class-imbalanced Target GDA.
>
> **A3:** Thank you for your insightful comment. **We conduct additional experiments** where the target domain is also processed to follow a long-tailed distribution using the same imbalance factor $\rho$ as the source domain. To provide a fair comparison, we selected the two most competitive baselines(TAM, GraphSHA) from our main experiments and report the results in the following table:
>
>
> For TAM：
> $\rho$|A2C|A2D|C2D|
> -|-|-|-|
> 10| $\downarrow$ 7.16%| $\downarrow$ 6.99%| $\downarrow$ 14.01%
> 20|$\downarrow$ 9.65%| $\downarrow$ 11.69%| $\downarrow$ 18.54%
> 50|$\downarrow$ 11.69%| $\downarrow$ 18.16%| $\downarrow$ 21.36%
>
> For GraphSHA:
> $\rho$|A2C|A2D|C2D|
> -|-|-|-|
> 10| $\downarrow$ 6.65%| $\downarrow$ 8.00%| $\downarrow$ 5.43%
> 20| $\downarrow$9.77%| $\downarrow$8.20%| $\downarrow$8.14%
> 50| $\downarrow$13.08%| $\downarrow$20.42%| $\downarrow$9.82%
>
> For ImGDA：
> $\rho$|A2C|A2D|C2D|
> -|-|-|-|
> 10|$\downarrow$**1.64%**|$\downarrow$**5.14%**|$\downarrow$**4.35%**
> 20|$\downarrow$**6.52%**|$\downarrow$**4.58%**|$\downarrow$**6.68%**
> 50|$\downarrow$**0.30%**|$\downarrow$**1.36%**|$\downarrow$**4.37%**
>
> In each cell of the table, the value indicates the **percentage decrease** in micro score compared to our experimental results (with balanced target domain in Table 1) under the imbalanced target domain setting.
>
> As shown in the table above, class imbalance in the target domain indeed leads to a noticeable performance drop for the models, particularly as the imbalance factor increases. In contrast, our method **maintains relatively stable performance** under the same conditions. This demonstrates that our approach not only effectively addresses the imbalance in the source domain but also exhibits strong robustness when faced with imbalance in the target domain. We speculate that this robustness stems from the contrastive learning component in our framework, which encourages the model to learn more stable and semantically consistent node representations, thereby mitigating the adverse effects of distributional bias in the target domain.
>
> In fact, in the experimental settings, we consider a scenario where the source domain exhibits a long-tailed label distribution while the target domain remains relatively balanced. Detailed statistics regarding the data distributions are provided in Appendix G. In the current body of research on GDA—including unsupervised GDA, semi-supervised GDA, and open-set GDA—most experimental setups assume that both the source and target graphs follow balanced distributions. In contrast, our work is **the first** to explicitly investigate the GDA task under the assumption that the source domain exhibits data imbalance, without imposing assumptions on the target domain distribution. This is **motivated by practical considerations:** source domain data is typically labeled and easier to obtain, making it feasible to observe and address long-tailed distributions; whereas for the target domain, due to the absence of labels, it is difficult to verify whether it is balanced or long-tailed.
>
> We hope that the additional experiments and analysis provide satisfactory answers to your concerns and demonstrate the robustness and stability of our method under long-tailed target domain settings.
> > **We hope our responses above can address your concerns and demonstrate the reliability and effectiveness of our proposed method**

---

> > ### Comment · Reviewer_XCp7 · 2025-08-03
> > **Comment on A1**
> >
> > In A1, the author defines the imbalance factor; however, this metric is sensitive to outlier classes. For example, even if the overall class distribution appears balanced, a single class with an extremely small or large sample size can disproportionately affect the result. To ensure a more reliable assessment of class imbalance, more robust metrics—such as the Gini coefficient or alternatives [1][2]—should be considered.
> >
> > In A3, the authors explain that the performance of ImGDA, TAM, and GraphSHA degrades to varying degrees under imbalanced target domain settings. While this overall trend is expected, one specific observation remains puzzling: as the imbalance factor increases, both TAM and GraphSHA show decreasing performance, whereas ImGDA exhibits an upward trend. This appears counter-intuitive. Could the authors clarify the underlying reason for this behavior?
> > As the author has already provided a clear and well-reasoned explanation of this issue in the response to reviewer MqCZ, further elaboration here is deemed unnecessary.
> >
> >
> > [1] Yang L, Jiang H, Song Q, et al. A survey on long-tailed visual recognition[J]. International Journal of Computer Vision, 2022, 130(7): 1837-1872.
> > [2] Cui Y, Jia M, Lin T Y, et al. Class-balanced loss based on effective number of samples[C]//Proceedings of the IEEE/CVF conference on computer vision and pattern recognition. 2019: 9268-9277.

---

> ### Author Response · Authors · 2025-08-03
>
> Thank you for your feedback and for suggesting the inclusion of additional imbalance metrics. For each imbalance factor selected in our paper, we provide the corresponding **Standard Deviation ($\sigma$)**, **Mean/Median Ratio ($\gamma$)**, and **Gini Coefficient ($\delta$)** [1], as shown in the tables below.
>
>
> $\rho=5$|$\sigma$|$\gamma$|$\delta$|
> -|-|-|-|
> ACMv9|628.79|1.17|0.31
> Citationv1|525.18|1.17|0.31
> DBLPv7|420.95|1.17|0.31
>
> $\rho=10$|$\sigma$|$\gamma$|$\delta$|
> -|-|-|-|
> ACMv9|714.46|1.36|0.41
> Citationv1|596.31|1.36|0.41
> DBLPv7|471.26|1.36|0.41
>
> $\rho=20$|$\sigma$|$\gamma$|$\delta$|
> -|-|-|-|
> ACMv9|762.39|1.66|0.49
> Citationv1|636.40|1.66|0.49
> DBLPv7|502.90|1.66|0.49
>
> $\rho=50$|$\sigma$|$\gamma$|$\delta$|
> -|-|-|-|
> ACMv9|798.08|2.25|0.57
> Citationv1|666.23|2.25|0.57
> DBLPv7|526.27|2.25|0.57
>
> $\rho=100$|$\sigma$|$\gamma$|$\delta$|
> -|-|-|-|
> ACMv9|814.47|2.92|0.62
> Citationv1|679.81|2.92|0.62
> DBLPv7|537.29|2.92|0.62
>
> From the tables, we observe that for a fixed imbalance factor, $\gamma$ and $\delta$ remain consistent across different datasets, while $\sigma$ varies. When the imbalance factor reaches 20 or higher, $\delta$ approaches or exceeds 0.5, indicating a significant long-tailed distribution in the dataset. This demonstrates that our **imbalance factor effectively captures the long-tailed nature of the data.** Thank you again for your suggestion. We also commit to incorporating some of these metrics for measuring data imbalance into the experimental descriptions in the revised version to make our study more rigorous.
>
> We hope our additional explanations can address your concerns.
>
> [1] Yang L, Jiang H, Song Q, et al. A survey on long-tailed visual recognition[J]. International Journal of Computer Vision, 2022, 130(7): 1837-1872.

---

> ### Comment · Reviewer_XCp7 · 2025-08-05
>
> The author has addressed my concerns to a reasonable extent.

---

### Official Review · Reviewer_RspL · 2025-06-30

**Clarity:** 3
**Significance:** 2
**Originality:** 3
**Rating:** 4
**Confidence:** 3

**Summary:**

This paper presents ImGDA, a dual-branch prototype-enhanced contrastive framework for unsupervised class-imbalanced graph domain adaptation that captures both local and global structure to generate class-specific prototypes from a distilled anchor set. It leverages cross-branch prototype contrast to rebalance and align class semantics within the source graph, and cross-domain prototype contrast—with non-parametric pseudo-labels and an adaptive temperature mechanism—to reduce distribution shifts and mitigate imbalance during training. Extensive experiments across multiple real-world graph datasets show that ImGDA consistently outperforms SOTA domain adaptation and imbalanced learning baselines.

**Questions:**

See Weaknesses.

**Ethical Concerns:**

["NO or VERY MINOR ethics concerns only"]

**Final Justification:**

The rebuttal clarifies to some extent, and I have adjusted the score.

**Limitations:**

See Weaknesses.

**Quality:**

3

**Strengths And Weaknesses:**

Strengths:
- The proposed setting that integrates class imbalance with unsupervised graph domain adaptation is novel and addresses real-world distribution shifts overlooked by prior work.
-	The dual-branch prototype-enhanced contrastive framework is effective at capturing both local and global semantics to rebalance source classes and align representations across domains.
-	Extensive experiments on multiple real-world datasets demonstrate ImGDA’s consistent superiority over SOTA baselines.

Weaknesses:
-	The authors only compare against standalone domain‐adaptation and standalone imbalance methods—intuitively, such methods perform poorly on tasks that combine domain shift with class imbalance—without evaluating combinations of domain‐adaptation GNNs with mature imbalance techniques, many of which can be integrated via simple plug-in loss functions or sampling strategies without any architectural changes.
-	Some recent baselines are missing, including the graph domain adaptation methods Pair-Align [1] and TDSS [2], as well as the graph imbalance learning method ConsisGAD [3].
-	Although the paper reports empirical efficiency metrics for ImGDA, it lacks a formal time and space complexity analysis to clarify how runtime and memory scale with graph size, class count, and anchor set.

[1] Shikun Liu, Deyu Zou, Han Zhao, and Pan Li. "Pairwise Alignment Improves Graph Domain Adaptation." ICML 2024.
[2] Wei Chen, Guo Ye, Yakun Wang, Zhao Zhang, Libang Zhang, Daixin Wang, Zhiqiang Zhang, and Fuzhen Zhuang. "Smoothness really matters: A simple yet effective approach for unsupervised graph domain adaptation." AAAI 2025.
[3] Nan Chen, Zemin Liu, Bryan Hooi, Bingsheng He, Rizal Fathony, Jun Hu, and Jia Chen. "Consistency training with learnable data augmentation for graph anomaly detection with limited supervision." In The Twelfth International Conference on Learning Representations. 2024.

---

> ### Author Rebuttal · Authors · 2025-07-28
>
> We appreciate your thoughtful comments and suggestions. We will make every effort to address your concerns as follows.
> > Q1: Additional Baselines.
>
> **A1:** Thanks for the comment! We appreciate your attention to our experimental design and insightful suggestion to combine domain adaptation GNNs with established long-tail learning methods. Additionally, we **conduct experiments incorporating several GDA methods combined with imbalance techniques and recently proposed domain adaptation methods** to assess their performance under our task setting.
>
> We choose GDA-SpecReg[1] as the baseline GDA method and combine it with either upsampling or loss function modification to compare their performance against our proposed method. The results are shown in the table below.
>
> For GDA-SpecReg + modify loss functions:
> $\rho$|A2C|A2D|C2D
> |-|-|-|-|
> 10|70.69+-2.44/67.56+-2.86|69.43+-1.98/63.43+-2.08|72.53+-2.18/67.32+-1.54
> 20|67.44+-2.34/63.59+-2.32|65.78+-2.41/59.32+-2.89|67.42+-2.68/63.02+-1.96
> 50|65.47+-2.68/58.46+-2.48|63.42+-2.38/56.35+-2.14|65.54+-2.24/60.78+-2.33
>
> For GDA-SpecReg + upsampling:
> $\rho$|A2C|A2D|C2D
> |-|-|-|-|
> 10|68.31+-1.04/66.31+-1.26|67.36+-1.28/61.42+-1.04|70.44+-1.46/65.41+-0.98
> 20|65.23+-1.48/62.61+-1.12|63.44+-1.89/60.28+-1.96|66.54+-1.38/62.68+-1.36
> 50|60.27+-1.46/55.74+-1.35|59.86+-2.03/54.33+-2.04|63.16+-1.64/58.69+-2.06
>
> For ImGDA:
> $\rho$|A2C|A2D|C2D
> |-|-|-|-|
> 10|77.93+-1.03/73.98+-2.74|71.34+-2.61/66.82+-1.82|73.06+-2.16/69.62+-2.98
> 20|77.30+-1.05/73.92+-0.83|71.23+-1.59/65.66+-1.25|68.48+-1.55/63.42+-2.03
> 50|73.53+-1.40/67.42+-1.23|69.14+-1.63/61.27+-1.50|70.57+-1.19/66.32+-1.86
>
> Each cell in the table presents both micro and macro scores. From the results, we can observe that simply integrating GDA with imbalance-handling techniques does improve performance on this task. However, as the imbalance factor increases, the performance of these models drops significantly. Among them, models based on upsampling perform worse than those using loss modification. We speculate that this is because upsampling methods heavily rely on the quality of the sampled data. Under severe data imbalance, the number of reference samples for tail classes is limited or overly homogeneous, which may lead to overfitting on those classes and thus result in poor generalization to the target domain. In contrast, our method **significantly outperforms both strategies,** clearly demonstrating that the proposed dual-branch contrastive learning framework achieves robust and consistent performance across various degrees of data imbalance.
>
> Meanwhile, we choose Pair-Align[2], TDSS[3], ConsisGAD[4] as additional state-of-the-art GDA methods and class-imbalanced methods to evaluate their performance under our task setting, and the results are shown below.
>
> For Pair-Align:
> $\rho$|A2C|A2D|C2D
> |-|-|-|-|
> 10|62.34+-0.18/56.87+-1.12|58.36+-3.46/53.74+-1.28|60.49+-1.63/58.44+-0.98
> 20|54.37+-1.24/49.54+-2.36|54.32+-1.08/48.24+-0.86|56.48+-2.14/52.34+-0.32
> 50|48.56+-2.10/35.32+-1.84|49.53+-3.46/43.43+-3.21|50.12+-1.65/42.42+-0.38
>
>
> For TDSS:
> $\rho$|A2C|A2D|C2D
> |-|-|-|-|
> 10|66.46+-0.48/62.42+-0.32|64.64+-1.26/59.89+-2.26|65.74+-0.43/61.48+-0.80
> 20|60.48+-0.87/56.43+-1.45|58.43+-2.28/50.36+-0.94|58.32+-1.12/54.36+-2.52
> 50|52.56+-1.26/43.46+-1.48|50.52+-2.36/42.03+-1.32|52.32+-3.84/43.74+-2.37
>
> For ConsisGAD:
> $\rho$|A2C|A2D|C2D
> |-|-|-|-|
> 10|72.48+-1.44/66.39+-0.32|68.43+-2.02/64.44+-1.28|69.98+-1.42/64.32+-1.45
> 20|66.34+-2.36/60.39+-1.98|64.56+-2.39/55.37+-2.44|64.43+-2.52/60.45+-1.30
> 50|62.32+-2.45/53.64+-2.04|60.86+-2.33/52.68+-2.59|58.33+-2.48/52.37+-1.28
>
> For ImGDA:
> $\rho$|A2C|A2D|C2D
> |-|-|-|-|
> 10|77.93+-1.03/73.98+-2.74|71.34+-2.61/66.82+-1.82|73.06+-2.16/69.62+-2.98
> 20|77.30+-1.05/73.92+-0.83|71.23+-1.59/65.66+-1.25|68.48+-1.55/63.42+-2.03
> 50|73.53+-1.40/67.42+-1.23|69.14+-1.63/61.27+-1.50|70.57+-1.19/66.32+-1.86
>
> Reviewing the three newly added baselines, we observe trends consistent with the conclusions in Section 5.2 of our main paper. Specifically, domain adaptation methods such as Pair-Align and TDSS perform poorly in our task setting, and their performance degrades significantly as the imbalance factor increases. Compared to other approaches, methods specifically designed to address class imbalance in graphs, such as ConsisGAD exhibits notably stronger performance. This highlights that source domain imbalance significantly hinders cross-domain knowledge transfer—potentially even more so than the domain discrepancy itself. Our method **outperforms** ConsisGAD by a substantial margin. As the imbalance factor increases, the performance of these baselines degrades considerably, whereas our method maintains relatively stable results. This clearly demonstrates the **effectiveness and robustness** of our approach in tackling the domain adaptation problem under source domain imbalance.
>
> To provide a more comprehensive evaluation of our method, we commit to including these baselines in the revised version. Thank you again for your valuable suggestion, and we hope these additional experiments adequately address the reviewer’s concerns.
>
> [1]You Y, Chen T, Wang Z, et al. Graph domain adaptation via theory-grounded spectral regularization[C]//The eleventh international conference on learning representations. 2023.
>
> [2] Shikun Liu, Deyu Zou, Han Zhao, and Pan Li. "Pairwise Alignment Improves Graph Domain Adaptation." ICML 2024.
>
> [3] Wei Chen, Guo Ye, Yakun Wang, Zhao Zhang, Libang Zhang, Daixin Wang, Zhiqiang Zhang, and Fuzhen Zhuang. "Smoothness really matters: A simple yet effective approach for unsupervised graph domain adaptation." AAAI 2025.
>
> [4] Nan Chen, Zemin Liu, Bryan Hooi, Bingsheng He, Rizal Fathony, Jun Hu, and Jia Chen. "Consistency training with learnable data augmentation for graph anomaly detection with limited supervision." In The Twelfth International Conference on Learning Representations. 2024.
>
> >Q2: Time Complexity and Space Complexity.
>
> **A2**: Thanks for the comment! We will provide the time complexity and space complexity below.
> For the time complexity, let $N^s$ and $|\mathcal{E}^s|$ denote the number of nodes and edges in the source domain graph, and $|N^t|$ and $|\mathcal{E}^t|$ denote the number of nodes and edges in the target domain graph. Assuming an $L$-layer GCN encoder with a feature dimension of $d$, the computational complexity of feature encoding is $\mathcal{O}(L|\mathcal{E}|^sd + LN^sd^2)$ for the source domain, and $\mathcal{O}(L|\mathcal{E}|^td + LN^td^2)$ for the target domain. When $|\mathcal{E}|^s \gg n$, this simplifies to $\mathcal{O}(|\mathcal{E}|^sd)$ and $\mathcal{O}(|\mathcal{E}|^td)$, respectively. For anchor node sampling, we sort nodes based on their degree and select $m$ nodes per class. Since $m \ll N^s$, the complexity is $\mathcal{O}(N^s \log N^s)$. For prototype computation, the Mean prototype computation per class takes $\mathcal{O}(md)$, the Cross-branch prototype contrastive loss takes $\mathcal{O}(md)$, the Cross-domain prototype contrastive loss takes $\mathcal{O}(N^td)$, and the Supervised loss computation in the source domain takes $\mathcal{O}(N^sd)$. Thus, the total time complexity is $\mathcal{O}(|\mathcal{E}^s|d + |\mathcal{E}^t|d + N^s \log N^s + md + N^td + N^sd)$
> When $|\mathcal{E}^s| \gg N^s \log N^s$, computation is dominated by the number of edges. When $|\mathcal{E}^s| \approx N^s\log N^s)$, sorting complexity becomes a key factor. If $N^t$ is very large, computation is significantly affected by the target domain nodes. As a result, the overall complexity can be approximated as:$\mathcal{O}(\max(|\mathcal{E}^s| + |\mathcal{E}^t|, N^s\log N^s, N^td))$. Besides, we have a detailed discussion of the time complexity of our method and several baselines in the appendix, please refer to it for more details.
>
> As for the space complexity: storing node features incurs a complexity of $\mathcal{O}(Nd)$; storing the sparse adjacency matrix requires $\mathcal{O}(\mathcal{|E|})$; the PPMI matrix has a space complexity of $\mathcal{O}(N^2)$; the model parameters require $\mathcal{O}( D^2)$; and storing the prototypes and pseudo-labels requires $\mathcal{O}(Cd)$ and $\mathcal{O}(NC)$ respectively, and C denotes the number of classes. In summary, the main space cost of our method arises from the PPMI matrix, which is $\mathcal{O}(N^2)$.
>
> > **We hope our responses above can address your concerns and show the stability and effectiveness of our proposed method**

---

> > ### Comment · Reviewer_RspL · 2025-08-05
> >
> > The rebuttal clarifies to some extent, and I have adjusted the score.

---

### Official Review · Reviewer_Gadc · 2025-06-30

**Clarity:** 2
**Significance:** 2
**Originality:** 1
**Rating:** 4
**Confidence:** 3

**Summary:**

The paper addresses an important problem of class imbalance in unsupervised graph domain adaptation (GDA). The proposed ImGDA framework integrates dual-branch encoders, prototype construction, and contrastive learning to mitigate source-domain bias and domain shifts. While the approach is technically sound and well-structured, significant concerns regarding motivation, novelty, and experimental design limit its contribution.

**Questions:**

See Weaknesses.

**Ethical Concerns:**

["NO or VERY MINOR ethics concerns only"]

**Final Justification:**

Thank you for your detailed and thoughtful rebuttal. I will raise my score.

**Limitations:**

yes

**Quality:**

2

**Strengths And Weaknesses:**

Strengths

1. Clear Formulation: The problem setup (Section 3) and modular framework (Section 4) are well-presented.

2. Theoretical Grounding: The theoretical analyses of the prototype contrastive loss (Theorems 4.1–4.2) provide a solid foundation.

Weaknesses

1. Insufficient Motivation and Empirical Analysis

The claim that class imbalance "inevitably leads to knowledge extraction bias" in GDA lacks robust empirical validation. The authors rely on a synthetic example (Figure 1) without demonstrating the prevalence of this issue in real-world graph datasets. Recommend to provide a thorough analysis of class imbalance in established graph benchmarks to quantify its impact on existing GDA methods. Statistical evidence would strengthen the motivation.

2. Limited Methodological Novelty

Prototype-based contrastive learning and anchor-based graph construction are well-studied in non-graph domains (e.g., CV/NLP) and recent graph learning. The paper does not clarify how ImGDA fundamentally advances these concepts for graphs. No graph-specific innovations (e.g., handling heterophily, noisy edges) are discussed. Recommend to explicitly differentiate ImGDA from prior work:
How do the "prototype-enhanced" losses uniquely address graph structural biases?

3. Weak Experimental Design

Synthetic Imbalance: Artificially inducing imbalance on balanced datasets (ACM/DBLP) does not reflect real-world scenarios. Performance on naturally imbalanced graphs (e.g., social networks with skewed communities) remains unverified.

Small-Scale Datasets: Experiments are limited to citation graphs. Larger heterogeneous graphs[1] can be also used in graph domain adaption task and should be included to test scalability and generalizability.

[1] Jiang X, Jia T, Fang Y, et al. Pre-training on large-scale heterogeneous graph[C]//Proceedings of the 27th ACM SIGKDD conference on knowledge discovery & data mining. 2021: 756-766.

---

> ### Author Rebuttal · Authors · 2025-07-28
>
> Thank you for your insightful comments, and we will try our best to address your concerns below.
> >Q1: Empirical Analysis about the Class-imbalanced Real-world Graph Datasets.
>
> **A1**: Thanks for your valuable comment! We appreciate the reviewer’s concerns regarding the motivation of our proposed scenario. In fact, multiple real-world scenarios exist where the source domain exhibits class imbalance with available labels, while the target domain remains entirely unlabeled. These scenarios have been under-explored, making our proposed approach both novel and significant.
>
> For instance, in citation networks, early citation networks serve as the source domain with abundant labels, while later citation networks constitute the unlabeled target domain. Due to the limited research in certain emerging fields during earlier periods, severe label imbalance exists. For instance, in the **arXiv citation network(which can be found in the OGB websites)**, articles are categorized into 40 fields. In the early citation network (spanning **1990–2010**), some fields contain as few as **six articles**, while the most populated field has **3,122 articles**, resulting in an extreme imbalance factor of **520**. And we give the specific statistical data of arXiv citation networks in the table below:
>
> Datasets|0|1|2|3|4|5|6|7|8|9|
> |-|-|-|-|-|-|-|-|-|-|-|
> arxiv[1970-2010]|63|36|887|122|403|485|147|71|619|650
>
> Datasets|10|11|12|13|14|15|16|17|18|19|
> |-|-|-|-|-|-|-|-|-|-|-|
> arxiv[1970-2010]|1134|83|9|197|137|52|200|25|**6**|9
>
> Datasets|20|21|22|23|24|25|26|27|28|29|
> |-|-|-|-|-|-|-|-|-|-|-|
> arxiv[1970-2010]|331|63|283|247|455|12|37|143|**3122**|33
>
> Datasets|30|31|32|33|34|35|36|37|38|39|
> |-|-|-|-|-|-|-|-|-|-|-|
> arxiv[1970-2010]|399|182|33|116|935|24|384|289|202|349
>
>
> The numbers 0 to 39 in the table denote the fields of articles, and the second row of each table denotes the number of the article. Similarly, in social network analysis, influential or highly active users (often a minority group) provide labeled data that is inherently imbalanced compared to the broader, unlabeled user base. In financial fraud detection, confirmed fraud cases (which are relatively few) serve as labeled source data, while the majority of transactions in emerging markets remain unlabeled, necessitating domain adaptation techniques for effective fraud detection.
> These examples demonstrate that domain adaptation in settings where the source domain is imbalanced and the target domain is unlabeled is not only relevant but also practically significant, particularly when leveraging graph-structured relationships through Graph Neural Networks (GNNs).
> >Q2: Explanation of the Methodological Novelty.
>
> **A2**: Thanks for the comment! Our work introduces a **new problem setting**: **graph domain adaptation (GDA) with class imbalance in the source domain**. To address this, we integrate GNNs with contrastive learning in a **dual-branch contrastive framework, which consists of a local branch and a global branch.** The local branch focuses on capturing local structural features from the graph using its original adjacency matrix, while the global branch incorporates high-level topological information via a PPMI matrix constructed from random walks, which captures more comprehensive global structural connectivity. Unlike prior works that directly apply prototype-based contrastive learning or anchor-based prototype construction, we embed these strategies into our dual-branch design, **enabling the consistent integration of both local and global structural information** throughout the entire model pipeline. This constitutes a graph-specific solution rather than a generic adaptation. Besides, our method inherently incorporates a mechanism that **alleviates graph structural biases through its dual-branch design.** Specifically, both the source domain imbalance handling and the cross-domain knowledge alignment are conducted within a dual-branch framework that leverages two different views—each constructed from a distinct adjacency matrix. One view retains the original graph structure, while the other captures a more global structural perspective. By applying cross-branch prototype contrastive learning based on these two views in the source domain, our method effectively **introduces structural perturbations** and learns semantically consistent node representations under such perturbations. This approach enhances the robustness of learned semantic information, **making it more resilient to structural shifts in the target domain graph.** Consequently, it helps to stabilize cross-domain semantic transfer and improves overall knowledge transfer performance.
>
> In summary, our work is **the first** to propose the problem of domain adaptation under source graph imbalance and **introduces a graph-specific dual-branch contrastive learning framework.** This design **mitigates the impact of graph structural biases by introducing structural perturbations** in the source domain, which makes the GNNs more resilient to structural shifts in the target domain, thus providing a promising solution in this regard. We appreciate your insightful suggestion and will consider explicitly addressing graph structure bias as a future research direction. This will be included in the revised version under the "Future Work" section to further advance research in imbalanced graph domain adaptation. We hope this explanation addresses your concerns regarding the novelty of our method.
> > Q3: More Experimental Results.
>
> **A3:** Thanks for the comment! As mentioned above, we have identified the OGB Arxiv citation graph as a naturally imbalanced dataset, and now we conduct additional experiments on this dataset, with Arxiv papers from 1970–2010 as the source domain and papers from 2012–2014 as the target domain. Meanwhile, we also compare our method with two strong baselines identified in the main manuscript. The experimental results are shown in the table below:
>
> Methods|arxiv[1970-2010]->arxiv[2012-2014]
> |-|-|
> ImGDA|42.34+-4.32/33.43+-2.26
> TAM|33.21+-6.89/25.31+-5.6
> GraphSHA|36.84+-3.24/23.45+-3.89
>
> In the table above, each cell presents micro-F1 / macro-F1. As shown, our method consistently **outperforms baselines on both metrics.** However, all methods yield relatively low macro-F1 scores, which is expected due to the inherent imbalance in the target domain (Arxiv 2012–2014). Macro-F1 is more sensitive to performance on minority classes, and the long-tail nature of the data exacerbates this sensitivity.
>
> Additionally, to evaluate our method on large-scale graphs, we conduct experiments on the CS dataset from the Open Academic Graph (OAG) [1]. Since OAG is a heterogeneous graph while our method is designed for homogeneous settings, we extract a homogeneous subgraph from the CS domain by using P-P (paper-paper) connections as edges, and treat "field" as the node label, with the remaining features used as attributes. We use the 2014 CS snapshot as the source domain and the 2017 snapshot as the target domain for node-level field classification. Due to the massive size of this dataset, we adopt a batching strategy for training and inference. We include two additional baselines for comparison. The micro-F1 results are presented below:
>
> $\rho$|ImGDA|TAM|GraphSHA
> |-|-|-|-|
> 10|32.25|18.47|16.62
> 20|25.64|12.84|11.92
> 50|20.68|10.97|9.83
>
>
> In this table, we observe that the performance of all methods degrades significantly on ultra-large-scale graphs. As the graph size increases, so does its complexity in terms of the number and diversity of nodes, placing substantial demands on GNN scalability and learning stability. Despite this, our method still shows a clear advantage over baselines, highlighting its robustness in handling large graphs.
>
> In summary, these additional experiments demonstrate the scalability and practical applicability of our method in both real-world imbalanced scenarios and extremely large graphs. We hope these results address your concerns and further validate the effectiveness of our proposed approach.
>
> [1] Jiang X, Jia T, Fang Y, et al. Pre-training on large-scale heterogeneous graph[C]//Proceedings of the 27th ACM SIGKDD conference on knowledge discovery & data mining. 2021: 756-766.
>
> > **Thank you again for your thoughtful comments. We hope our response can address your concerns and demonstrate the effectiveness and validity of our approach.**

---

> > ### Comment · Area_Chair_37vw · 2025-08-05
> > **Author-Reviewer Discussion Reminder**
> >
> > Dear Reviewer Gadc,
> >
> > As the deadline for author-reviewer discussion is approaching, could you please check the authors' rebuttal and post your response?
> >
> > Thank you!
> >
> > Best,
> >
> > AC

---

> > > ### Comment · Reviewer_Gadc · 2025-08-07
> > >
> > > Thank you for the reminder. I have carefully read the authors’ rebuttal and find that it resolves most of my original concerns. The authors have provided extensive clarification and additional experiments, especially on naturally imbalanced and large-scale datasets. I have added a follow-up question regarding runtime comparison to better assess the method's practicality.

---

> > ### Comment · Reviewer_Gadc · 2025-08-07
> >
> > Thank you for your detailed and thoughtful rebuttal. I appreciate the comprehensive empirical evidence provided for real-world class imbalance, as well as the clarification on the dual-branch framework and its graph-specific design. These additions have addressed most of my initial concerns. That said, since the paper also highlights the time-consuming nature of existing approaches, and additional experiments were conducted on large-scale datasets like arXiv, I would encourage the authors to provide a comparison of time complexity or training/inference time between ImGDA and the baselines. This would further strengthen the paper's practicality and scalability claims.

---

> > > ### Author Response · Authors · 2025-08-07
> > >
> > > Thank you for your feedback and suggestions. Below, we provide the time complexity analysis of our method:
> > >
> > > Let $N^s$ and $|\mathcal{E}^s|$ denote the number of nodes and edges in the source domain graph, and $N^t$ and $|\mathcal{E}^t|$ denote the number of nodes and edges in the target domain graph. Assuming an $L$-layer GCN encoder with a feature dimension of $d$, the computational complexity of feature encoding is $\mathcal{O}(L|\mathcal{E}|^sd + LN^sd^2)$ for the source domain, and $\mathcal{O}(L|\mathcal{E}|^td + LN^td^2)$ for the target domain. When $|\mathcal{E}|^s \gg n$, this simplifies to $\mathcal{O}(|\mathcal{E}|^sd)$ and $\mathcal{O}(|\mathcal{E}|^td)$, respectively. For anchor node sampling, we sort nodes based on their degree and select $m$ nodes per class. Since $m \ll N^s$, the complexity is $\mathcal{O}(N^s \log N^s)$. For prototype computation, the Mean prototype computation per class takes $\mathcal{O}(md)$, the Cross-branch prototype contrastive loss takes $\mathcal{O}(md)$, the Cross-domain prototype contrastive loss takes $\mathcal{O}(N^td)$, and the Supervised loss computation in the source domain takes $\mathcal{O}(N^sd)$. Thus, the total time complexity is $\mathcal{O}(|\mathcal{E}^s|d + |\mathcal{E}^t|d + N^s \log N^s + md + N^td + N^sd)$
> > > When $|\mathcal{E}^s| \gg N^s \log N^s$, computation is dominated by the number of edges. When $|\mathcal{E}^s| \approx N^s\log N^s)$, sorting complexity becomes a key factor. If $N^t$ is very large, computation is significantly affected by the target domain nodes. As a result, the overall complexity can be approximated as:$\mathcal{O}(\max(|\mathcal{E}^s| + |\mathcal{E}^t|, N^s\log N^s, N^td))$.
> > >
> > > In addition, we present a comparison and analysis of the runtime efficiency between our method and the baselines in the Appendix, which shows that our method maintains comparable running time while achieving better performance. Please refer to Appendix D.4 for more details.
> > >
> > > We hope our response addresses your questions.

---

> > > ### Author Response · Authors · 2025-08-09
> > >
> > > Dear Reviewer Gadc,
> > >
> > > Thank you for your recognition of our rebuttal and your further feedback. We have provided detailed responses regarding your concerns about the comparison of runtime efficiency, and we would like to know if our responses have addressed your concerns? As the rebuttal and discussion phase is coming to an end, we truly value the opportunity to respond to reviewers’ questions and warmly welcome any additional feedback or inquiries. We firmly believe this will help us refine and improve our work.
> > >
> > > Thank you again for the time and effort you have invested in enhancing our work.
> > >
> > > Best regards,
> > >
> > > The Authors

---

### Official Review · Reviewer_MqCZ · 2025-07-03

**Clarity:** 2
**Significance:** 3
**Originality:** 3
**Rating:** 5
**Confidence:** 3

**Summary:**

This paper addresses the problem of graph unsupervised domain adaptation and aims to address the problem of class imbalance, which has been overlooked by prior methods. To overcome this, the authors propose the Dual Prototype-Enhanced Contrastive Framework (ImGDA). ImGDA consists of (1) dual-branch embeddings that simultaneously learn local and global structure information, (2) anchor-based prototype construction that considers the balance between classes, and (3) prototype-aware contrastive learning that simultaneously performs class alignment and domain alignment, and aims to mitigate both class imbalance and domain difference problems.

**Questions:**

See the weakness section.

**Ethical Concerns:**

["NO or VERY MINOR ethics concerns only"]

**Final Justification:**

I appreciate the author's thoughtful and detailed rebuttal. The reviewer's response clarified the concerns I had raised. Therefore, I will increase my score.

**Limitations:**

The limitations of the proposed method are described in the Appendix. E (Broader Impact and Limitations).

**Paper Formatting Concerns:**

No major formatting issues were found. The paper appears to follow the NeurIPS 2025 formatting guidelines.

**Quality:**

3

**Strengths And Weaknesses:**

Strengths
1. The performance is very good overall, and it is especially impressive that the performance remains stable for a large imbalance factor ρ, with a large performance gap to the baseline.
2. Visualizations and various ablation experiments are used to intuitively demonstrate the effectiveness of the proposed method.

Weaknesses
1. In Figure 2 (Main Framework), there is no alignment between the caption and the visual elements, making it difficult to intuitively understand which parts of the figure are referred to by the components described in the text.
2. Section 4.3 claims that aligning prototypes between local and global branches contributes to performance improvements, but it lacks an explanation of how this alignment relates to mitigating the class imbalance problem. It would be better to see some rationale or intuitive interpretation of why simply aligning the two branches improves the tail class representation.
3. In Figure 4, the w/o local and w/o global settings seem to remove one of the two branches required for computing L_{cb}; wouldn't this make the use of L_{cb} infeasible in those cases?  It is somewhat surprising that these settings outperform the w/o L_{cb}, and require further explanation of experimental conditions.
4. While the degree of class imbalance in the source domain is clearly controlled by the imbalance factor ρ, it is not specified whether the target domain has a balanced or equally unbalanced distribution. To reflect a more practical scenario, the paper would benefit from a discussion or empirical results on whether the proposed method remains effective when the target domain is also class-imbalanced.

---

> ### Author Rebuttal · Authors · 2025-07-28
>
> We sincerely appreciate your comments and valuable suggestions, and we will give detailed responses to your concerns below.
>
> >Q1: The Alignment between the Caption and the Visual Elements in the Main Framework Figure 2.
>
> **A1:** Thanks for the comment! Our main framework consists of three core components: Dual-Branch Embedding Generalization, Anchor-Based Prototype Construction, and Prototype-Enhanced Contrastive Learning. As shown in Figure 2, the left side illustrates the Dual-Branch Embedding Generalization module using global and local views to capture richer graph representations.  The upper-right and lower-right parts of Figure 2  represent the Anchor-Based Prototype Construction and Prototype-Enhanced Contrastive Learning modules. After selecting anchor nodes from the source domain, we perform cross-branch and cross-domain prototype contrastive learning, which simultaneously mitigates the issue of class imbalance in the source domain and facilitates effective cross-domain knowledge transfer. The contents of the framework are consistent with the textual description, and **we appreciate your highly constructive suggestion — we promise we will revise the figure in the revised version** to explicitly mark each component, aligning them more clearly with the figure caption to improve reader comprehension. We sincerely appreciate your thoughtful feedback.
>
> >Q2: Interpretation of Why Aligning Two Branches could Improve the Tail Class Representation.
>
> **A2:** Thanks for comments! In Section 4.2, we **theoretically** explain how the anchor-based prototype construction changes the lower bound of the supervised contrastive loss, shifting it from a head-class-dominated regime to a more balanced one. **This adjustment reduces the adverse impact of head classes,** which usually dominate the source domain, on the learning of other class semantics.
>
> Additionally, we introduce a **dynamic temperature** in the contrastive loss, which is computed based on the class frequency. This temperature plays two roles: it penalizes hard negative samples and modulates the gradient magnitude for each class. For head classes, a larger temperature increases attention to hard negatives while reducing the gradient, thus lowering their influence. For tail classes, a smaller temperature has a limited effect on hard negative focus (due to the scarcity of samples) but increases their gradient, amplifying their contribution to the loss. As a result, **this mechanism effectively reduces head-class dominance while promoting tail-class learning, alleviating the data imbalance.**
>
> The purpose of Cross-Branch Prototype Contrastive Learning is to align local and global semantic views, allowing the model to capture more class-representative features while addressing imbalance. In summary, the alignment of semantic information across branches integrates multiple techniques to jointly mitigate class imbalance and enhance the quality of learned representations. We hope this response clarifies your concerns.
> >Q3: Further Explanation of Ablation Study Experiments.
>
> **A3:** Thank you for your detailed comment and your attention to our method. In the ablation study, **w/o local** and **w/o global** refer to replacing the original local and global dual-branch prototype contrastive learning setup with global and global and local and local configurations, respectively. In such cases, **the cross-branch prototype contrastive learning essentially degenerates into single-branch prototype contrastive learning.**
>
> Our motivation for designing the dual-branch prototype contrastive learning is to fully leverage both the local and global information of the graph. When the dual branches are replaced with a single branch, the absence of the complementary view leads to the loss of additional semantic information, resulting in performance degradation. **However, compared to completely removing this module (i.e., w/o $\mathcal{L}_{cb}$), the single-branch settings still allow the model to benefit from at least one semantic view, thus preserving some consistency in the learned class features.** Therefore, w/o local and w/o global perform better than w/o $\mathcal{L}_{cb}$. We promise we will include a detailed explanation of this setting in the revised version to help readers better understand our experimental setup.
> >Q4: Discussion of the Imbalanced Distribution in the  Target Domain.
>
> **A4:** Thanks for comments! **We conduct additional experiments** where the target domain is also processed to follow a long-tailed distribution using the same imbalance factor $\rho$ as the source domain. To provide a fair comparison, we selected the two most competitive baselines (TAM, GraphSHA) from our main experiments and report the results in the following table:
>
>
> For TAM：
> $\rho$|A2C|A2D|C2D|
> -|-|-|-|
> 10| $\downarrow$ 7.16%| $\downarrow$ 6.99%| $\downarrow$ 14.01%
> 20|$\downarrow$ 9.65%| $\downarrow$ 11.69%| $\downarrow$ 18.54%
> 50|$\downarrow$ 11.69%| $\downarrow$ 18.16%| $\downarrow$ 21.36%
>
> For GraphSHA:
> $\rho$|A2C|A2D|C2D|
> -|-|-|-|
> 10| $\downarrow$ 6.65%| $\downarrow$ 8.00%| $\downarrow$ 5.43%
> 20| $\downarrow$9.77%| $\downarrow$8.20%| $\downarrow$8.14%
> 50| $\downarrow$13.08%| $\downarrow$20.42%| $\downarrow$9.82%
>
> For ImGDA：
> $\rho$|A2C|A2D|C2D|
> -|-|-|-|
> 10|$\downarrow$**1.64%**|$\downarrow$**5.14%**|$\downarrow$**4.35%**
> 20|$\downarrow$**6.52%**|$\downarrow$**4.58%**|$\downarrow$**6.68%**
> 50|$\downarrow$**0.30%**|$\downarrow$**1.36%**|$\downarrow$**4.37%**
>
> In each cell of the table, the value indicates the **percentage decrease** in micro score compared to our experimental results (with balanced target domain in Table 1) under the imbalanced target domain setting.
>
> As shown in the table above, class imbalance in the target domain indeed leads to a noticeable performance drop for the models, particularly as the imbalance factor increases. In contrast, our method **maintains relatively stable performance** under the same conditions. This demonstrates that our approach not only effectively addresses the imbalance in the source domain but also exhibits strong robustness when faced with imbalance in the target domain. We speculate that this robustness stems from the contrastive learning component in our framework, which encourages the model to learn more stable and semantically consistent node representations, thereby mitigating the adverse effects of distributional bias in the target domain.
>
> In fact, in the experimental settings, we consider a scenario where the source domain exhibits a long-tailed label distribution while the target domain remains relatively balanced. Detailed statistics regarding the data distributions are provided in Appendix G. In the current body of research on GDA—including unsupervised GDA, semi-supervised GDA, and open-set GDA—most experimental setups assume that both the source and target graphs follow balanced distributions. In contrast, our work is **the first** to explicitly investigate the GDA task under the assumption that the source domain exhibits data imbalance, without imposing assumptions on the target domain distribution. This is **motivated by practical considerations:** source domain data is typically labeled and easier to obtain, making it feasible to observe and address long-tailed distributions; whereas for the target domain, due to the absence of labels, it is difficult to verify whether it is balanced or long-tailed.
>
> We hope these additional experiments and analyses help to address your concerns.
> > **Thank you again for your suggestions. We hope the responses above could address your concerns and highlight the practical significance and robustness of our work.**

---

> > ### Comment · Reviewer_XCp7 · 2025-08-03
> > **Comment on A4**
> >
> > In A4, the authors explain that the performance of ImGDA, TAM, and GraphSHA degrades to varying degrees under imbalanced target domain settings. While this overall trend is expected, one specific observation remains puzzling: as the imbalance factor increases, both TAM and GraphSHA show decreasing performance, whereas ImGDA exhibits an upward trend. This appears counter-intuitive. Could the authors clarify the underlying reason for this behavior?

---

> > > ### Author Response · Authors · 2025-08-03
> > >
> > > Thanks for your feedback and careful observation. We also noticed this phenomenon during our experiments, and we would like to share our findings with you.
> > >
> > > In the additional experiments we conducted, the domain adaptation part in our method would undoubtedly be affected by the long-tailed distribution target domain.  However, **a key observation** is that as the class imbalance in the target domain becomes more severe, the overall accuracy on the target domain becomes increasingly dependent on the head classes, as the tail classes contribute less and less to the overall accuracy due to their extremely small quantity. Therefore, the final performance can **follow two competing trends**:
> > > >**1. A decrease in performance due to degraded cross-domain knowledge transfer caused by the imbalance in pseudo-label quality;**
> > >
> > > >**2. An apparent performance improvement (or a smaller drop) as the number of tail-class nodes becomes negligible, thus reducing their impact on the overall accuracy.**
> > >
> > > And **the final performance depends on which of the two trends dominates.** In our experimental results, the two comparison methods (TAM and GraphSHA) are mainly designed to address class imbalance and fail to transfer knowledge across domains.  As a result, the first trend dominates, leading to noticeable performance drops. In contrast, our method mitigates both class imbalance and domain shift, effectively alleviating the impact of the first trend and thereby outperforming the baselines.
> > >
> > > Moreover, **for the cases the reviewer mentioned** in which performance increases with higher imbalance factors. We hypothesize this may be because **the tail classes become so rare that their influence on the micro score is minimal, allowing the second trend to dominate.** As a result, performance may even improve compared to cases with milder imbalance. To further validate our hypothesis, we conduct experiments with $\rho=100$. The results are shown below.
> > >
> > > $\rho=100$|A2C|A2D|C2D|
> > > -|-|-|-|
> > > balanced $\mathcal{G}^t$|69.30|68.61|64.74
> > > imbalanced $\mathcal{G}^t$|75.33|73.43|69.80
> > >
> > > As the results show, compared to balanced target domain, the results not only do not decrease but even improve when the target domain is highly imbalanced. This confirms our hypothesis: as the imbalance factor increases, the second trend increasingly dominates, leading to less performance drop or even performance improvement.
> > >
> > >
> > > We hope this explanation addresses your concern.

---

> > ### Comment · Reviewer_MqCZ · 2025-08-05
> >
> > I appreciate the author's thoughtful and detailed rebuttal. The reviewer's response clarified the concerns I had raised. Therefore, I will increase my score.

---

### Author Response · Authors · 2025-08-09
**Summary of Rebuttal**

Dear Area Chairs and Reviewers,

We sincerely appreciate the time and effort the AC dedicated during the discussion period to help us address the reviewers’ concerns. We are also deeply grateful to the reviewers for their thoughtful and insightful comments, which have been invaluable in improving the quality of our paper.

In summary, our paper has received positive comments like **"well-presented"**(reviewer Gadc), and our work also gets recognitions like **"especially impressive" "demonstrate the effectiveness"** (reviewer MqCZ), **"novel"** (reviewer RspL), **"innovative with a well-organized and coherent structure"** (reviewer XCp7), which has **"solid theoretical foundation"** (reviewer Gadc). Besides, our explanations during the rebuttal and discussion period gained reviewers' feedback like **"thoughtful and detailed"** (reviewer MqCZ, Gadc), which addresses the reviewers' concerns (reviewer MqCZ, Gadc, RspL, XCp7).

During the rebuttal and discussion period, we make every effort to address the reviewers' concerns, which can be concluded as follows:

>- We conducted experiments with **additional real-world arxiv datasets.**
>- We conducted experiments with **additional advanced methods.**
>- We conducted experiments with **imbalanced target domain.**
>- We supplied a **thorough analysis** of the counterintuitive phenomenon in the additional experiments with imbalanced target domain.
>- We supplied **more explanations for the details** (novelty, imbalance factor, pseudo-labels) of our method.
>- We supplied **time complexity and space complexity analysis** of our method.
>- We supplied a **detailed table** about a real-world imbalanced datasets.

We believe that our timely responses have addressed most of the reviewers’ questions. Therefore, we are confident that our work is innovative, theoretically grounded, and experimentally validated, making a meaningful contribution to the field of graph machine learning, especially in graph domain adaptation. Moreover, we will incorporate the improvements gained from both the rebuttal and discussion period into our revised version. With these refinements, we believe our work will become even more solid and robust.

Once again, we sincerely thank you for taking the time to help us improve our work.

Best regards,

The Authors

---

### Note · Authors · 2025-08-12

Dear Area Chairs and Reviewers,

We would like to express our sincere gratitude to the program committee for granting us the opportunity to provide final remarks, as well as to all Area Chairs and Reviewers for their efforts in reviewing our work and offering constructive suggestions.

In our work, we are **the first** to introduce the problem of graph domain adaptation under **source-domain class imbalance** scenario and propose a **graph-specific dual-branch contrastive learning framework** to tackle this challenge. Specifically, our method leverages two branches to capture both local and global information of the graph, followed by cross-branch and cross-domain prototype contrastive learning with dynamic temperature. Consequently, the framework simultaneously mitigates the class imbalance in the source domain and enables effective cross-domain knowledge transfer. We also provide the **theoretical guarantee** and experimental analysis to further demonstrate the effectiveness of our framework.

In summary, we have received positive feedback from reviewers, and the comments can be listed as:

>- The paper is **"well-presented"**. (Reviewer Gadc)
>- The work is **"especially impressive"** and **"demonstrate the effectiveness"**. (Reviewer MqCZ)
>- The method is **"novel"** and **"innovative with a well-organized and coherent structure"**, with **"solid theoretical foundation"**. (Reviewer RspL, XCp7, Gadc)
>- The rebuttal is **"thoughtful and detailed"**. (Reviewer MqCZ, Gadc)

During the rebuttal and discussion period, we carefully considered the reviewers’ concerns and feedback. Through our response, we believe that we have **addressed most of the reviewers’ concerns**.

We believe that our work has the potential to contribute to the graph domain adaptation communities, which also holds strong potential for adaptation to a wide range of domains beyond the current setting. Once again, we sincerely thank the reviewers and ACs for your dedication and insightful feedback, which have truly enriched and strengthened our work. We promise to include all the improvements during the rebuttal in the final version of the paper.

Best regards,

The Authors

---

### Decision · Program_Chairs · 2025-09-17

**Decision:**

Accept (poster)

**Comment:**

This paper aims to address the class imbalance problem in graph unsupervised domain adaptation. The authors proposed a dual-branch prototype-enhanced contrastive framework named ImGDA. It leverages cross-branch prototype contrast to rebalance and align class semantics within the source graph, and uses cross-domain prototype contrast to reduce distribution shifts and mitigate imbalance during training. Experiments on multiple real-world graph datasets demonstrate the effectiveness of the proposed approach.

All of the reviewers recognized the novelty and technical contributions of this work. The problem setting is well motivated and clearly presented. The theoretical analyses of the prototype contrastive loss provide a solid foundation. Also, it is impressive that the performance of the proposed approach remains stable for a large imbalance factor. Overall, the paper is well written and easy to follow.

Meanwhile, reviewers raised some questions regarding novelty, technical details, baselines, time complexity, etc. The authors have provided detailed responses with additional results, which have addressed the previous concerns from reviewers. The authors are strongly encouraged to incorporate the new results and discussions into the final version of the paper.